# PhysicsMinions: Winning Gold Medals in the Latest Physics Olympiads with a Coevolutionary Multimodal Multi-Agent System

## Abstract

Physics is central to understanding and shaping the real world, and the ability to solve physics problems is a key indicator of real-world physical intelligence. Physics Olympiads, renowned as the crown of competitive physics, provide a rigorous testbed requiring complex reasoning and deep multimodal understanding, yet they remain largely underexplored in AI research. Existing approaches are predominantly single-model based, and open-source MLLMs rarely reach gold-medal-level performance. To address this gap, we propose PhysicsMinions, a coevolutionary multi-agent system for Physics Olympiad. Its architecture features three synergistic studios: a Visual Studio to interpret diagrams, a Logic Studio to formulate solutions, and a Review Studio to perform dual-stage verification. The system coevolves through an iterative refinement loop where feedback from the Review Studio continuously guides the Logic Studio, enabling the system to self-correct and converge towards the ground truth. Evaluated on the HiPhO benchmark spanning 7 latest physics Olympiads, PhysicsMinions delivers three major breakthroughs: **(i) Strong generalization:** it consistently improves both open-source and closed-source models of different sizes, delivering clear benefits over their single-model baselines; **(ii) Historic breakthroughs:** it elevates open-source models from only 1–2 to 6 gold medals across 7 Olympiads, achieving the first-ever open-source gold medal in the latest International Physics Olympiad (IPhO) under the average-score metric; and **(iii) Scaling to human expert:** it further advances the open-source Pass@32 score to 26.8/30 points on the latest IPhO, ranking 4th of 406 contestants and far surpassing the top single-model score of 22.7 (ranked 22nd). Generally, PhysicsMinions offers a generalizable framework for Olympiad-level problem solving, with the potential to extend across disciplines.

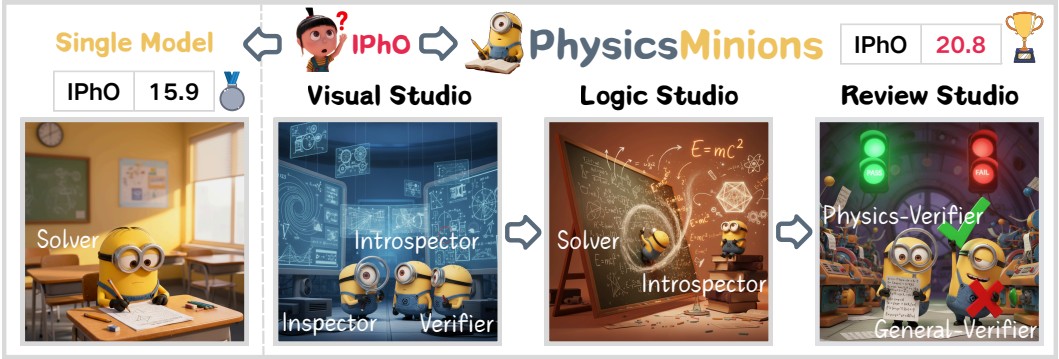

Figure 1: Illustration of PhysicsMinions, a coevolutionary multimodal multi-agent system. It comprises three studios: the *Visual Studio* for visual extraction, the *Logic Studio* for solution generation and refinement, and the *Review Studio* for dual verification. Each agent in each studio likes a "Minion"—while an individual Minion is limited, their collaboration forms a coevolutionary system for tackling Olympiad-level problems. For example, in the latest IPhO, Intern-S1 alone scores only 15.9 (silver), whereas with PhysicsMinions it reaches 20.8, achieving a gold medal.

## 1 INTRODUCTION

A deep understanding of physics is essential for shaping the real world, and the ability to solve physics problems is a critical step toward developing real-world physical intelligence (Zheng et al., 2025; Dai et al., 2025). Physics Olympiads stand out as the crown of competitive problem solving, requiring both complex physics reasoning and advanced multimodal understanding. Yet results from the HiPhO benchmark (Yu et al., 2025), dedicated to physics Olympiads, expose the limitations of current single-model paradigms. On the latest International Physics Olympiad (IPhO), only three closed-source models barely surpassed the gold medal threshold, while no open-source MLLM achieved gold, with most scoring near or below the bronze cutoff. These outcomes highlight both the formidable difficulty of Olympiad-level physics and the limitation of single-model approaches.

Encouragingly, advances in multi-agent systems have demonstrated the potential of agent-driven reasoning (Madaan et al., 2023; Wang et al., 2025b). At the latest International Mathematical Olympiad (IMO), leading single models such as GPT-5 and Gemini-2.5-Pro scored well below the bronze medal line[1] , yet with a multi-agent framework they solved 5 of 6 problems, reaching gold-level performance (Huang & Yang, 2025). This shows the promise of multi-agent paradigms in overcoming reasoning bottlenecks. However, unlike the purely text-based IMO, physics Olympiads present unique challenges: **(1) Complex physical reasoning**, involving equation derivations, applications of laws and theorems, and long-horizon dependencies; **(2) Multimodal understanding**, as figures, plots, and diagrams often contain indispensable information. These challenges make existing mathematical multi-agent frameworks insufficient for direct application to physics reasoning.

To address this gap, we present **PHYSICSMINIONS**, the pioneering coevolutionary multimodal multi-agent system tailored for physics Olympiads. As illustrated in Fig. 1, each agent acts like a "Minion": a single one may be unable to solve Olympiad-level problems alone, but through coevolutionary collaboration, the system achieves complex reasoning. PHYSICSMINIONS is organized into three specialized studios: **(1) Visual Studio**, which transforms the visual inputs into structured information; **(2) Logic Studio**, where a solver generates an initial solution and an introspector iteratively improves it; **(3) Review Studio**, which employs both a physics-specific and a general verifier for dual-stage checking. The three studios interact in a coevolutionary loop: Visual Studio validates and refines extracted information, Review Studio feeds verification results back to guide correction, and Logic Studio integrates these signals to iteratively refine solutions, enabling the system to substantially enhance multimodal physical reasoning and progressively approach the ground truth.

Evaluations on the HiPhO benchmark (Yu et al., 2025), spanning 7 latest physics Olympiads, validate the effectiveness of PHYSICSMINIONS with three key breakthroughs: **(1) Strong generalization:** the system consistently improves both closed- and open-source models with different scales, outperforming their single-model baselines; **(2) Historic breakthroughs:** open-source models that achieved only 1-2 gold medals alone now obtain 6 in 7 Olympiads, including the first-ever gold in the latest IPhO; **(3) Scaling to human expert:** the open-source Pass@32 score on IPhO reaches 26.8/30 points, placing 4$^{th}$ among 406 contestants and well above the top single-model result of 22.7 (22$^{nd}$ place). These show that PHYSICSMINIONS elevates MLLMs to gold-medal performance and toward human-expert levels, highlighting its potential as a general framework for problem solving.

Our work makes the following contributions:

- **A new paradigm for Olympiad-level physics reasoning.** We introduce PHYSICSMINIONS, a pioneering coevolutionary multimodal multi-agent system tailored for physics Olympiads. Unlike the single-model paradigm, our system leverages coevolutionary agents and cross-modal information integration to push beyond the single-model ceiling on the latest physics Olympiads.

- **Coevolutionary framework design.** We develop a coevolutionary framework with three specialized studios, where the *Visual Studio* validates and refines visual information, the *Review Studio* performs dual-stage verification to guide refinement in the *Logic Studio*. All three interact in a coevolutionary loop, thereby strengthening multimodal physical reasoning.

- **Historic breakthroughs.** Our framework consistently improves both closed- and open-source models, raising open-source results from 1-2 to 6 gold medals across 7 Olympiads and delivering the first-ever open-source gold in the latest IPhO. It further achieves 4$^{th}$ place among 406 contestants with Pass@32 score via open-source Intern-S1, surpassing 99% contestants.

---

[1]See results at `https://matharena.ai/?comp=imo--imo_2025`.

## 2 RELATED WORK

**Challenges in Multimodal Physics Olympiads.** Benchmarks such as PhysUniBench (Wang et al., 2025a) and SeePhys (Xiang et al., 2025) highlight the difficulty of combining physics reasoning with visual information in multimodal problems. OlympiadBench (He et al., 2024) and OlympicArena (Huang et al., 2024) show the formidable complexity of physics Olympiad problems requiring long-horizon reasoning. Building on both, HiPhO (Yu et al., 2025) (High School Physics Olympiad Benchmark) poses dual challenges from multimodal understanding and Olympiad-level physics reasoning, where evaluations reveal that open-source MLLMs rarely achieve gold-medal performance, underscoring the difficulty of multimodal physical reasoning at the Olympiad level.

**Reasoning Methods and Multi-agent Frameworks.** Various methods have been developed to enhance the reasoning capabilities of (M)LLMs. Single-model approaches include Chain-of-Thought (Wei et al., 2022) to strengthen step-by-step reasoning, Best-of-N (Stiennon et al., 2020) to select the most reliable answer from multiple attempts, and Self-MoA (Li et al., 2025) to aggregate diverse outputs through varied prompting. More recently, multi-agent frameworks such as Self-Refine (Madaan et al., 2023) and Generative Self-Refinement (Wang et al., 2025b) adopt self-reflection, enabling models to improve their own solutions. Mathematical multi-agent systems have also gained attention, exemplified by the framework proposed for IMO (Huang & Yang, 2025), which applies self-verification and self-improvement to pure-text mathematical problems; however, such methods cannot be directly extended to the multimodal challenges of physics Olympiads.

## 3 PHYSICSMINIONS: A COEVOLUTIONARY MULTI-AGENT SYSTEM

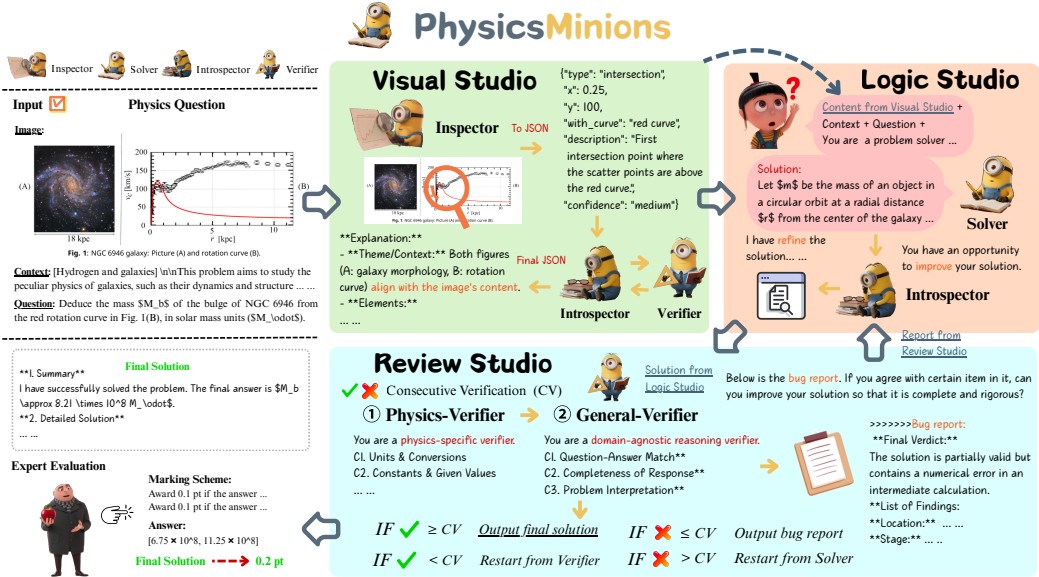

Figure 2: Overview of PHYSICSMINIONS, a coevolutionary multimodal multi-agent system. Given a multimodal problem, the *Visual Studio* extracts structured visual information. The *Logic Studio* generates an initial solution and improves it. The *Review Studio* then conducts dual-stage verification; failures trigger bug reports returned to the *Logic Studio* for further revision. This loop continues until the solution passes consecutive checks, forming the coevolutionary process.

### 3.1 OVERVIEW OF FRAMEWORK

PHYSICSMINIONS consists of three coevolutionary studios: the *Visual Studio*, the *Logic Studio*, and the *Review Studio*. Given a multimodal problem with diagrams or plots, the Visual Studio first observes, validates, and reflects on the input to extract structured information, which is then passed to the Logic Studio. In the Logic Studio, a solver generates an initial solution and an introspector

refines it through self-improvement before passing it on. The Review Studio then applies dual-stage verification: the Physics-Verifier checks physical consistency (e.g., constants and units), while the General-Verifier conducts more detailed inspections of logic, reasoning, and calculations. If either stage fails, a detailed bug report is returned to the Logic Studio, where the introspector revises the solution and resubmits it to Review Studio for verification. This process repeats until the solution passes a predefined number of consecutive verifications (CV), which is the only hyperparameter in the system. A solution that passes CV checks consecutively is accepted as the final solution; if it fails CV times consecutively, the solver regenerates a new candidate solution. This collaborative critique-and-refine cycle defines the system's coevolutionary process, with CV set to 2 by default (see Section 4.5 for analysis). We next provide a detailed introduction of each studio.

### 3.2 VISUAL STUDIO

**Pipeline.** The Visual Studio consists of three cooperative agents: an *Inspector*, an *Introspector*, and a *Verifier*. Given a physics problem with multimodal diagrams, the Inspector first determines the image type (e.g., `plot`, `curve`, `free-body`, `circuit`, etc.) and then extracts task-relevant details. For a plot, it records axis labels, ranges, and tick values; for curves, it distinguishes colors and line styles, identifying features such as endpoints and peaks; for free-body diagrams, it lists objects together with the forces and their directions. These features are converted into a structured JSON description, which the Introspector refines to be self-contained, consistent, and faithful to the image. The Verifier then checks for errors such as wrong values, missing elements, or inconsistencies. If problems are found, a bug report is returned to the Introspector for revision before resubmission. Once the description passes a predefined number of consecutive verifications (CV), it is accepted as an accurate representation of the visual information. This structured output, rather than the raw image, is then passed to the Logic Studio along with the problem statement.

**Innovation.** Unlike prior methods that feed raw images directly into the model, the Visual Studio converts multimodal inputs into structured JSON through iterative observation, introspection, and verification. This process reduces ambiguity, ensures consistency, and bridges visual perception with symbolic reasoning. As shown in our ablation studies (Section 4.5), such validated and refined representations yield significantly better performance than raw images, underscoring the importance of structured visual information for complex physics reasoning.

### 3.3 LOGIC STUDIO

**Pipeline.** The Logic Studio consists of two cooperative agents: a *Solver* and an *Introspector*. Given the problem statement and the structured JSON from the Visual Studio, the Solver generates a *structured solution* with two components: **(i) a Summary**, which declares the solution as *Complete* or *Partial*, provides the final answer if complete (or rigorously proven results if partial), and outlines a method sketch; and **(ii) a Detailed Solution**, which presents a step-by-step analysis with all equations written in TEX. The Introspector then improves this solution, focusing on equation derivation, numerical calculations, and overall consistency. The revised solution is then passed to the Review Studio for dual-stage verification. If verification fails, the Introspector receives a bug report and revises the solution; when it disagrees with certain items, it provides explicit justifications to avoid repeated misunderstandings. This loop continues until the solution passes the required number of consecutive verifications (CV) or, after persistent failures, the Solver regenerates a new candidate.

**Innovation.** The Logic Studio enforces a *structured solution format* (Summary + Detailed Solution) that makes reasoning explicit and errors traceable, enabling targeted bug reports and precise refinements. Combined with verification feedback, the Solver–Introspector collaboration forms a critique-and-refine coevolution: solutions are iteratively corrected where they fail, reinforced where they hold, and progressively driven toward the ground truth.

### 3.4 REVIEW STUDIO

**Pipeline.** The Review Studio performs dual-stage verification with a *Physics-Verifier* followed by a *General-Verifier*. The Physics-Verifier first carries out domain-specific checks, including coarse validation of units and physical constants, as well as finer checks of assumptions and physical consistency (e.g., detecting when a formula is applied to the wrong type of quantity). If the solution

Table 1: Evaluation results on 7 latest physics Olympiads from the HiPhO benchmark using the exam score metric. Gold, Silver, and Bronze indicate scores above the medal thresholds, following HiPhO. Only the theoretical parts of the exams are considered, so Full Mark (Model) $\leq$ Full Mark (Human). Top-1 Score (Human) is the highest score among human medalists, while Top-1 Score (Model) is the best single-model score on the HiPhO leaderboard.

| Latest Physics Olympiads | IPhO | APhO | EuPhO | NBPhO | PanPhO | PanMechanics | F=MA | Medal |
|---|---|---|---|---|---|---|---|---|
| Full Mark (Human) | 30.0 | 30.0 | 30.0 | 72.0 | 100.0 | 100.0 | 25.0 | |
| Full Mark (Model) | 29.4 | 30.0 | 29.0 | 43.5 | 100.0 | 100.0 | 25.0 | |
| Top-1 Score (Human) | 29.2 | 30.0 | 27.0 | 53.2 | 81.0 | 62.0 | 25.0 | |
| Top-1 Score (Model) | 22.7 | 27.9 | 14.9 | 34.1 | 60.3 | 72.1 | 22.8 | |
| Gold Medal | 19.7 | 23.3 | 16.5 | 28.6 | 41.5 | 52.0 | 15.0 | |
| Silver Medal | 12.1 | 18.7 | 9.8 | 20.1 | 28.5 | 36.0 | 11.0 | |
| Bronze Medal | 7.2 | 13.1 | 5.8 | 15.2 | 14.5 | 20.0 | 9.0 | |
| Gemini-2.5-Flash-Thinking | 20.2 | 27.4 | 13.2 | 29.0 | 44.6 | 60.5 | 17.8 | 6 1 0 |
| + PHYSICSMINIONS | 21.5 | 28.0 | 16.5 | 33.3 | 57.8 | 72.0 | 24.0 | 7 0 0 |
| Intern-S1 | 15.9 | 21.7 | 9.0 | 23.0 | 41.1 | 60.4 | 18.4 | 2 4 1 |
| + PHYSICSMINIONS | 20.8 | 25.2 | 10.1 | 28.9 | 46.8 | 68.7 | 22.7 | 6 1 0 |
| InternVL3.5-241B-A28B | 12.0 | 21.1 | 9.4 | 22.6 | 24.9 | 54.7 | 14.0 | 1 3 3 |
| + PHYSICSMINIONS | 20.9 | 24.6 | 9.8 | 29.6 | 46.2 | 66.7 | 21.0 | 6 1 0 |
| Qwen2.5VL-32B-Instruct | 9.9 | 16.5 | 6.9 | 15.3 | 22.5 | 28.1 | 7.6 | 0 0 6 |
| + PHYSICSMINIONS | 12.4 | 17.7 | 9.0 | 21.0 | 29.5 | 36.0 | 12.0 | 0 5 2 |

fails, a bullet-point bug report is generated and returned directly to the Logic Studio for refinement, bypassing the second stage. If it passes, the solution proceeds to the General-Verifier, which begins with coarse checks for completeness and problem understanding (e.g., missing sub-questions or mis-interpretations), followed by fine-grained step-by-step verification of logical consistency, reasoning flow, and algebraic or numerical calculations. Failures at this stage trigger a comprehensive bug report. This report is then returned to the Logic Studio, where the Introspector revises the solution until it passes both stages consecutively.

**Innovation.** The Review Studio introduces two key innovations. First, the dual-stage design separates physics-specific and general verification: the Physics-Verifier quickly filters out domain errors, while the General-Verifier ensures broader logical soundness across disciplines. Second, both verifiers adopt a coarse-to-fine strategy, starting with high-level checks before moving to detailed verification. This layered design improves error coverage, and the structured bug reports provide clear guidance for the Introspector, enabling efficient and precise corrections toward ground truth.

## 4 EXPERIMENTS

### 4.1 EXPERIMENTAL SETUP

**Evaluation.** We evaluate PHYSICSMINIONS on seven latest physics Olympiads from the HiPhO benchmark (Yu et al., 2025), which covers both international and regional competitions. Following the HiPhO setup, we fixed the model temperature at 0.6 and applied both answer-level and step-level evaluation based on the official marking schemes, enabling direct human-level comparison. For each exam, we record the average score across three repeated inference runs. Our code is available at https://anonymous.4open.science/r/PhysicsMinions.

**Models.** We evaluate four representative MLLMs, spanning closed- and open-source models at different scales. **(1) Gemini-2.5-Flash-Thinking** (Comanici et al., 2025), a closed-source model with strong reasoning ability, ranks 2nd overall on the HiPhO leaderboard. **(2) Intern-S1** (Bai et al., 2025a) is the top-ranked open-source MLLM on the HiPhO leaderboard. **(3) InternVL3.5-241B-A28B** (Wang et al., 2025c), a flagship open-source model, achieves state-of-the-art results across diverse benchmarks. **(4) Qwen2.5VL-32B-Instruct** (Bai et al., 2025b), a medium-scale open-source model, allows us to assess how PHYSICSMINIONS enhances models with more limited capacity.

## 4.2 Overall Breakthroughs with PhysicsMinions

The main results in Table 1 show that PHYSICSMINIONS achieves three major breakthroughs on 7 latest physics Olympiads, significantly advancing multimodal physical reasoning.

**(1) PHYSICSMINIONS delivers consistent improvements over single-model baselines.** The gains hold across both closed- and open-source models, regardless of architecture or scale. All four evaluated MLLMs achieve higher exam scores with the system, demonstrating its broad and reliable effectiveness. As illustrated in Fig. 3, PHYSICSMINIONS consistently boosts Intern-S1's scores across all problems on the latest IPhO, with the most notable improvement in the Text+Data Figure modality, underscoring its strong ability to enhance multimodal reasoning.

**(2) PHYSICSMINIONS enables open-source MLLMs to achieve substantial medal progression—from bronze to silver, and from silver to gold.** For instance, Qwen2.5VL-32B-Instruct, a relatively weaker reasoner, improved from zero silvers to five silvers. Intern-S1 advanced from 2 golds, 4 silvers, and 1 bronze in the single-model setting to 6 golds and 1 silver with PHYSIC-SMINIONS. Likewise, InternVL3.5-241B-A28B with PHYSICSMINIONS also reached 6 golds and 1 silver, with remarkable leaps from bronze to gold in IPhO and PanPhO. Notably, this work marks **the first time** that open-source MLLMs have achieved a gold medal in the latest IPhO, including both Intern-S1 and InternVL3.5-241B-A28B, underscoring how the coevolutionary system, through verification and reflection, can elevate open-source reasoning to the Olympiad gold level.

**(3) PHYSICSMINIONS pushes the boundary of the closed-source MLLM, surpassing the top single model and even human contestants in several Olympiads.** With PHYSICSMINIONS, Gemini-2.5-Flash-Thinking became the first model to win gold medals in all seven Olympiads. Notably, while no single model on the HiPhO leaderboard had ever reached gold in the latest EuPhO, the system accomplished this milestone. It also outperformed the best single-model scores on three exams—APhO, EuPhO, and F=MA. In mechanics, the gains were especially striking: the system scored 24/25 on F=MA (25 multiple-choice questions), nearly perfect, and even surpassed human contestants in PanMechanics, demonstrating unprecedented advances in mechanical reasoning.

In summary, PHYSICSMINIONS consistently boosts performance, enables medal breakthroughs for open-source MLLMs, and pushes closed-source models beyond prior limits, showcasing its potential for advancing multimodal physical reasoning at Olympiad scale.

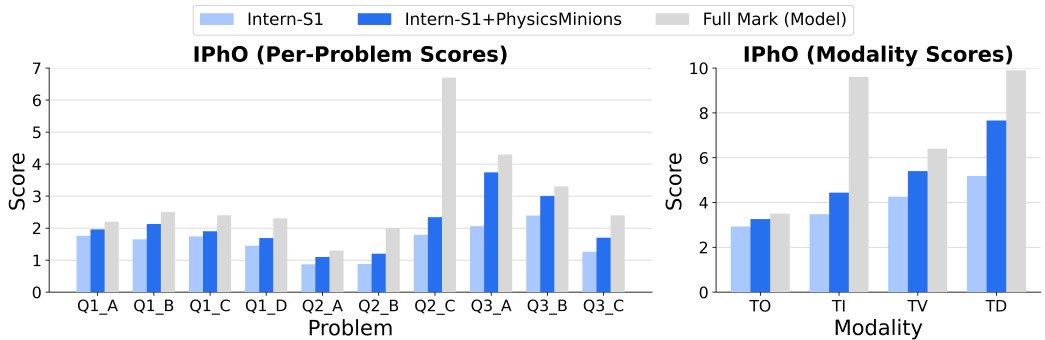

Figure 3: Performance improvement of Intern-S1 with PHYSICSMINIONS on the latest IPhO, shown by per-problem and modality scores. The HiPhO benchmark defines four modality types: TO = Text-Only, TI = Text+Illustration Figure, TV = Text+Variable Figure, and TD = Text+Data Figure.

## 4.3 Comparison with Other Frameworks

We compare PHYSICSMINIONS with three representative baselines: Best-of-N, Self-Mixture-of-Agents (Self-MoA), and Self-Refine. Best-of-N (Stiennon et al., 2020) selects the output with the highest total exam score from $N$ independent runs. Self-MoA (Li et al., 2025) ensembles diverse outputs from a single model by varying prompting strategies and then aggregates them into a final solution. Self-Refine (Madaan et al., 2023) iteratively prompts the model to critique and improve its own answers. Further details and discussion of these methods are provided in Appendix D.

Table 2: Performance comparison of different frameworks on 7 latest physics Olympiads using the exam score metric. Gold, Silver, and Bronze indicate scores above the respective thresholds. **Bold** marks the highest score, with PHYSICSMINIONS achieving the best results in all settings.

| Latest Physics Olympiads | IPhO | APhO | EuPhO | NBPhO | PanPhO | PanMechanics | F=MA | Medal | | |
|---|---|---|---|---|---|---|---|---|---|---|
| Gold Medal | 19.7 | 23.3 | 16.5 | 28.6 | 41.5 | 52.0 | 15.0 | 🥇 | | |
| Silver Medal | 12.1 | 18.7 | 9.8 | 20.1 | 28.5 | 36.0 | 11.0 | 🥈 | | |
| Bronze Medal | 7.2 | 13.1 | 5.8 | 15.2 | 14.5 | 20.0 | 9.0 | 🥉 | | |
| Intern-S1 | 15.9 | 21.7 | 9.0 | 23.0 | 41.1 | 60.4 | 18.4 | 2 | 4 | 1 |
| + **PHYSICSMINIONS** | **20.8** | **25.2** | **10.1** | **28.9** | **46.8** | **68.7** | **22.7** | 6 | 1 | 0 |
| + Best-of-N ($N = 3$) | 16.6 | 22.9 | 9.7 | 25.2 | 46.0 | 68.5 | 21.0 | 3 | 3 | 1 |
| + Self-MOA | 15.8 | 21.7 | 9.9 | 16.9 | 42.7 | 68.5 | 19.0 | 3 | 3 | 1 |
| + Self-Refine | 18.6 | 24.6 | 9.0 | 28.4 | 37.8 | 59.0 | 21.0 | 3 | 3 | 1 |

As shown in Table 2, PHYSICSMINIONS exhibits clear advantages in both consistent score improvement and medal progression. Its coevolutionary system enhances performance more reliably than alternative frameworks. In contrast, Self-MoA can produce incorrect solutions when ensembling candidates, as illustrated by Intern-S1 scoring lower than the single model in NBPhO. Self-Refine, while employing verification and reflection, lacks the dual-stage verification and coevolutionary reflection of PHYSICSMINIONS, resulting in lower scores in PanPhO and PanMechanics. In terms of medal outcomes, PHYSICSMINIONS elevates Intern-S1 from two to six golds, including at IPhO, whereas other frameworks achieve only three. Furthermore, PHYSICSMINIONS surpasses the Best-of-N performance, indicating it breaks through the performance ceiling of individual models.

## 4.4 SCALING PERFORMANCE UNDER PASS@$k$

Pass@$k$ evaluates a model's best score over $k$ independent attempts by taking the highest-scoring solution per problem. Fig. 4 illustrates the scaling behavior of PHYSICSMINIONS, revealing three key observations. **(1) Continuous performance evolution:** Intern-S1+PHYSICSMINIONS improves steadily from Pass@1 to Pass@4 with a 4.7-point gain, surpassing the top single-model score of 22.7 (22nd among humans). At Pass@32, it reaches 26.8/30, **ranking 4th of 406 contestants** and surpassing 99% of human contestants, with further growth potential. **(2) Early breakthroughs:** significant progress is achieved at Pass@4, where Intern-S1 upgrades from silver to gold, and Qwen2.5VL-32B-Instruct rises from bronze to silver. **(3) Base model determines ceiling:** the intrinsic capability of the base model constrains the system's upper bound. Stronger models, such as Intern-S1, benefit more from the coevolutionary system, yielding larger performance gains.

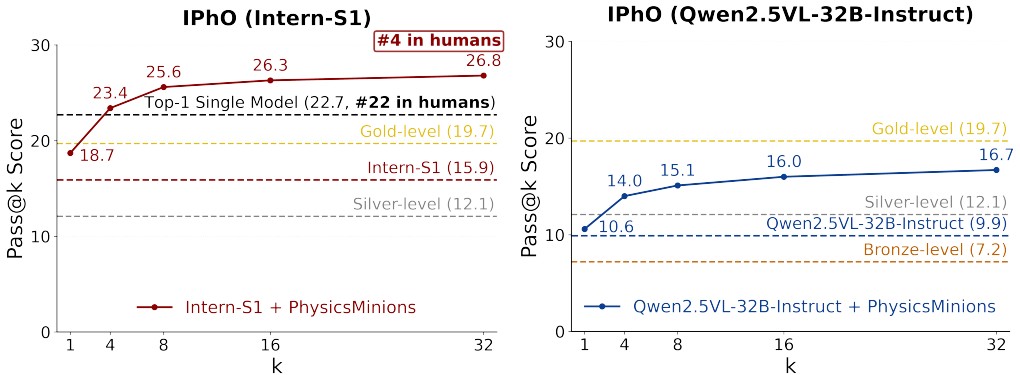

Figure 4: Scaling performance of Intern-S1 and Qwen2.5VL-32B-Instruct on the latest IPhO.

## 4.5 ABLATION STUDIES AND HYPERPARAMETER ANALYSIS

**Effect of Visual Studio.** Visual Studio converts diagrams and plots into structured JSON descriptions (see Fig. 2), which we compare against directly using raw images. As shown in Fig. 5(a),

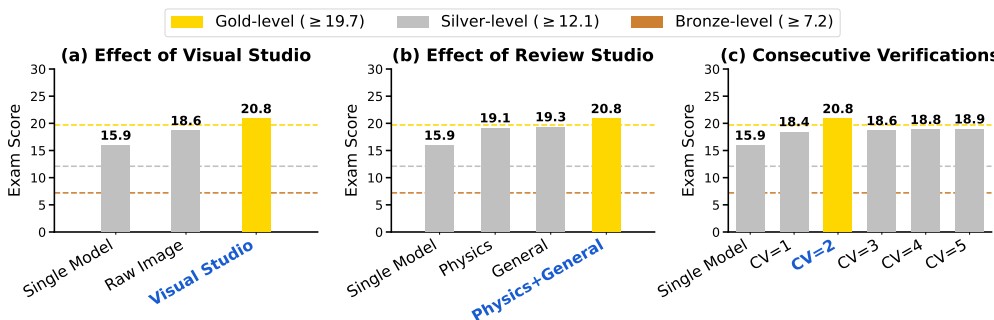

Figure 5: Ablation study and hyperparameter analysis using Intern-S1 on the latest IPhO.

Visual Studio achieves higher scores, since structured representations provide explicit cues that support reasoning. In contrast, raw images are harder for the solver to interpret directly, making it difficult to extract quantitative details and leading to weaker performance.

**Effect of Review Studio.** Review Studio employs a dual-stage verification with a Physics-Verifier followed by a General-Verifier. We compare each verifier used individually against their combination. As shown in Fig. 5(b), the combination yields the highest score of 20.8, whereas the Physics-Verifier alone only checks physics consistency and the General-Verifier alone lacks domain-specific rigor. Together, the two verifiers complement each other to achieve gold-level performance.

**Effect of Consecutive Verifications.** The key hyperparameter in the system is the number of consecutive verifications (CV), where a candidate solution is accepted only if it passes CV checks in a row. On the latest IPhO, we tested $CV \in \{1, 2, 3, 4, 5\}$ and observed the score rises from 15.9 to 20.8 at $CV = 2$, while larger values may trigger overthinking and reduce scores. As CV increases, token consumption increases accordingly, reaching about 1.9× and 3.2× that of $CV = 2$ for $CV = 3$ and $CV = 5$, respectively (measured on IPhO Q3-A6 as an example). Thus, we adopt $CV = 2$ as an empirically efficient setting, though the optimal value may vary with problem difficulty or model, with its effectiveness confirmed in our experiments.

## 5 DISCUSSION

### 5.1 CASE STUDY ON DUAL-STAGE VERIFICATION

As illustrated in Fig. 6, we present a case study showing how dual-stage verification refines solutions. **(1) The first stage:** The Physics-Verifier conducts domain-specific checks on units, conversions, and physical consistency, identifying that the characteristic time $\tau$ has an incorrect unit under dimensional analysis. Guided by this bug report, the introspector corrects the error by substituting the proper physical variable. **(2) The second stage:** The General-Verifier evaluates logical consistency, derivation soundness, and numerical validity. It detects a derivation error that a time-dependent $q_a$ leads to a flawed derivation of $n_c$. The introspector then re-derives $n_c$ and arrives at the correct solution with full credit. This case underscores the essential role of dual-stage verification in our coevolutionary system: the capacity to detect errors is the foundation of iterative self-improvement, and without reliable error detection, repeated reflection cannot converge on the correct answer.

### 5.2 LIMITATIONS

Visual Studio significantly improves multimodal reasoning, yet precise data extraction remains challenging. For instance, in Fig. 7, a chart with three peaks is partially misinterpreted: PHYSICSMINIONS correctly extracts two values, outperforming a single model, but still selects one peak incorrectly. Besides, we evaluated various chart analysis tools (Masry et al., 2024b;a; PaddlePaddleTeam, 2025; AiDescribePicture, 2024; Flux AI, 2024; Graph2Table, 2024; Image Describer X, 2025; Zhao et al., 2025; TextIn, 2025), most of which exhibit worse recognition capabilities than the single Gemini model, while only WebPlotDigitizer (Automeris, Inc., 2024) achieves fully correct extraction but requires human-in-the-loop operations, making it impractical for automated systems. These results underscore the need to develop fully automated, high-precision visual extraction capabilities to further enhance multimodal reasoning performance. Illustrations of tools are provided in Appendix F.

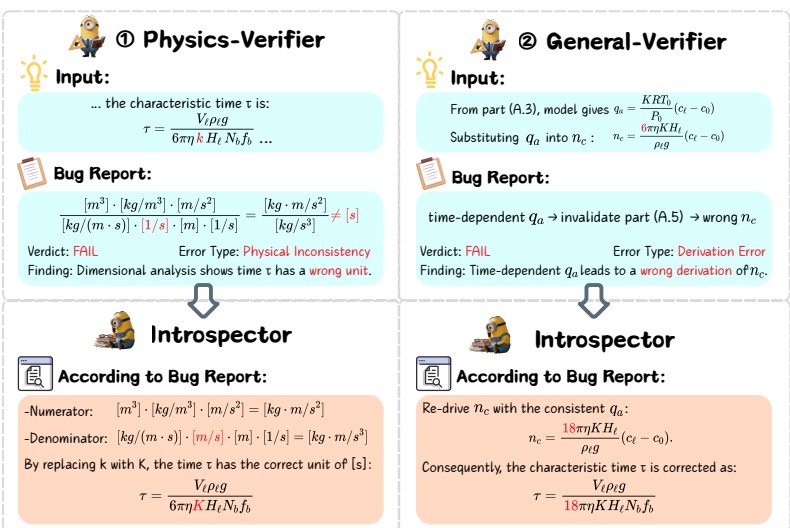

Figure 6: Case study of dual-stage verification on IPhO Q3-A6 using Intern-S1. The single model achieves only 0.2 points, whereas PHYSICSMINIONS, with dual-stage verification and self-reflection, attains the full score of 1.1 points, demonstrating substantial improvement.

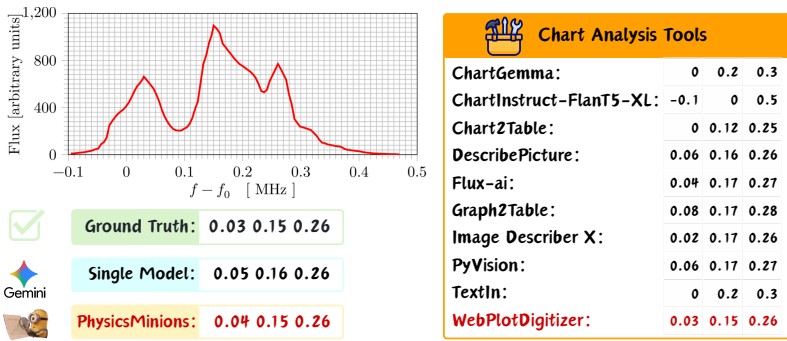

Figure 7: Limitation of PHYSICSMINIONS' Visual Studio in precise data extraction. Example: IPhO Q1-C4 requires identifying the x-coordinates of all three peaks in the curve.

## 5.3 RELATED WORK ON MLLM-BASED CHART AND FIGURE UNDERSTANDING

Recent work has begun to examine the ability of MLLMs to interpret charts and scientific figures, which is a critical skill for scientific reasoning. Studies such as CharXiv (Wang et al., 2024) and ECDBench (Yang et al., 2025) systematically benchmark this capability and show that even the most advanced MLLMs often struggle with chart understanding tasks, which is consistent with our observations. As shown in Fig. 7, Gemini-2.5-Flash-Thinking and most tools fail to produce reliable interpretations, especially on data figures. To address this, we propose the reflection-verification Visual Studio tailored to the visual demands of physics Olympiads. By iteratively validating and refining extracted visual information, the Visual Studio significantly improves figure interpretation, as further illustrated through case studies in Appendix G.

## 5.4 RELATED WORK ON MULTI-AGENT SYSTEM FOR PHYSICS OLYMPIADS

While our work and Physics Supernova (Qiu et al., 2025) are contemporaneous, our contributions differ substantially in three key aspects. **(1) Architecture:** Physics Supernova relies on the strong abilities of Gemini-2.5-Pro and external tools, whereas our system addresses the visual and verification limitations of most MLLMs through a reflection-verification enhanced image-reading module (Visual Studio) and a general-specialized integrated dual-verifier module (Review Studio), enabling robust reasoning even for open-source models. Our system focuses on improving the model's inter-

nal capabilities through coevolution rather than outsourcing reasoning. **(2) Generalization:** Physics Supernova focuses only on IPhO 2025 and strong-to-strong reasoning with Gemini-2.5-Pro, while our coevolutionary design provides consistent gains for both open- and closed-source models of different sizes across seven distinct Olympiads. **(3) Outcomes:** Our system enables weak-to-strong transitions (e.g., Intern-S1 achieving the first open-source gold on IPhO 2025) and further advances strong models (e.g., Gemini-2.5-Flash-Thinking reaching all golds), whereas Physics Supernova does not consider weaker models. These differences underscore the design innovation, broader generality, and applicability of our system.

## 5.5 RESOURCE CONSUMPTION COMPARISON BETWEEN SINGLE MODEL AND OUR SYSTEM

While our system introduces additional inference overhead compared to single-model baselines, it offers a clear advantage: it consistently breaks the performance ceiling of standalone models through reflection-verification cycles. On IPhO 2025, a single Intern-S1 run takes about 180 seconds and consumes 10,227 tokens per problem; our system requires 500 seconds and 89,482 tokens, which is roughly $3\times$ runtime and $9\times$ token usage. Under equivalent resource budgets, our system achieves avg@3 = 20.8, whereas the Best-of-N result reaches only 17.1 under equal runtime ($N = 9$) and equal token budget ($N = 27$), showing limited improvement. Moreover, our framework enables weaker models to perform comparably to stronger ones. For example, Intern-S1 improves from 2 gold medals to 6, matching Gemini-2.5-Flash-Thinking, offering a more lightweight and deployable solution under limited computational resources and deployment budgets.

## 6 GENERALIZATION ACROSS OTHER DISCIPLINES AND PHYSICS RESEARCH

Beyond physics Olympiads, we further evaluate the performance on other disciplines: (1) the mathematics competition AIME 2025, (2) two general science QA subset from the Scientists' First Exam (SFE-Astronomy and SFE-Earth) (Zhou et al., 2025), and (3) a frontier physics research benchmark CritPt (Zhu et al., 2025) that probes large model reasoning on scientific discovery tasks. As shown in Table 3, PHYSICSMINIONS consistently improves the performance of the base model Gemini-2.5-Flash-Thinking. Most notably, on the example challenge of CritPt, the system raises accuracy from 40% to 100% on Checkpoint 1 and enables breakthrough performance on Checkpoint 2, improving from 0% to 60% when provided with the answer from Checkpoint 1. These results demonstrate our system's potential for broader deployment in scientific reasoning and discovery.

Table 3: Accuracy improvements across other disciplines and physics research tasks.

| Benchmark | AIME 2025 | SFE Astronomy | SFE Earth | CritPt-Ckpt 1 | CritPt-Ckpt 2 (w/o Answer) | CritPt-Ckpt 2 (w/ Answer) |
|---|---|---|---|---|---|---|
| Gemini-2.5-Flash-Thinking | 78.3% | 24.3% | 32.0% | 40.0% | 0.0% | 0.0% |
| + PHYSICSMINIONS | 93.3% | 43.7% | 46.9% | 100.0% | 20.0% | 60.0% |
| *Improvement* | +15.0% | +19.4% | +14.9% | +60.0% | +20.0% | +60.0% |

## 7 CONCLUSION AND FUTURE WORK

Physics underpins our ability to shape the real world, and physics Olympiads distill this challenge into rigorous tests that expose the limits of single-model approaches. To address this, we proposed PHYSICSMINIONS, a coevolutionary multimodal multi-agent system designed to push beyond the single-model ceiling. By integrating a Visual Studio for structured perception, a Logic Studio for iterative solution refinement, and a Review Studio for dual-stage verification, the framework evolves solutions through continuous critique and feedback. Evaluated on 7 latest physics Olympiads, it delivers historic breakthroughs, including the first open-source gold in the latest IPhO and a Pass@32 score of 26.8/30 that ranks 4[th] among 406 contestants, surpassing 99% of human contestants. Furthermore, it shows consistent gains across other disciplines, highlighting its broad generality.

Future work will focus on three directions: (i) enhancing visual understanding and multimodal perception in the Visual Studio, (ii) expanding tool use with external solvers and domain-specific verifiers to further strengthen the reflection-verification cycle, and (iii) extending the coevolutionary paradigm to other disciplines, and potentially to real scientific research tasks.

ETHICS STATEMENT

All authors have read and adhered to the ICLR Code of Ethics. This work involves no human subjects, sensitive personal data, or potentially harmful applications. All datasets used in our experiments are publicly available from official sources, and no proprietary or private data are involved. The authors declare that there are no potential conflicts of interest related to this work.

REPRODUCIBILITY STATEMENT

To support reproducibility, we provide a complete anonymous codebase at `https://anonymous.4open.science/r/PhysicsMinions`, which is also included in the supplementary material. The repository contains the implementation of our proposed multimodal multi-agent system, the evaluation pipeline for the physics Olympiads, and the associated dataset, along with a detailed README that provides installation and usage instructions. Besides, Section 3 presents the architecture and workflow of the system, while Appendix C includes the prompts employed. Together, these resources enable reliable reproduction of our experiments and results.

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

# Supplemental Material of PHYSICSMINIONS

This document provides supplementary material to complement the main paper. It includes detailed descriptions of the system, prompts, comparative frameworks, additional results, and analysis tools. Specifically:

- **Appendix A** describes how large language models were used in this work.
- **Appendix B** presents a detailed description of the PHYSICSMINIONS system, including:
    - Appendix B.1: Image processing pipeline
    - Appendix B.2: Coevolutionary iteration strategy
    - Appendix B.3: Implementation details
- **Appendix C** provides the complete set of prompts used in PHYSICSMINIONS, including:
    - Appendix C.1: Prompts of Visual Studio
    - Appendix C.2: Prompts of Logic Studio
    - Appendix C.3: Prompts of Review Studio
- **Appendix D** introduces the comparative frameworks considered in our experiments, including:
    - Appendix D.1: Best-of-N strategy
    - Appendix D.2: Self-MoA
    - Appendix D.3: Self-Refine
    - Appendix D.4: Advantages over comparison frameworks
- **Appendix E** provides the description of physics Olympiads and additional results, including:
    - Appendix E.1: Overview of physics Olympiads
    - Appendix E.2: Performance gains across modality types
    - Appendix E.3: Performance gains across physics fields
- **Appendix F** gives an overview of the chart analysis tools tested in Section 5.2.

## A    THE USE OF LARGE LANGUAGE MODELS

In this work, the large language model GPT-5 was used as a general-purpose tool for polishing the writing, including improving clarity and grammar. In Fig. 1, the four images were generated with the assistance of Google Nano Banana[2], while the overall framework was created by the authors. The Minion-style icons in Fig. 2 were produced with the help of Doubao[3]. The conceptual design of both figures were entirely implemented by the authors. In addition, the evaluation pipeline follows the HiPhO benchmark (Yu et al., 2025), using Gemini-2.5-Flash as the grader. All prompts were initially designed by humans and lightly polished by GPT-5. No other substantive use of LLMs was involved in the ideation or methodology of this paper.

## B    A COEVOLUTIONARY SYSTEM OF PHYSICSMINIONS

### B.1    IMAGE PROCESSING PIPELINE

The system incorporates a dedicated image processing pipeline for handling image inputs:

1. **Image Reading:** Extracts initial visual information from the input image.
2. **Image Improvement:** Improves the interpretation of the extracted image content.
3. **Image Verification:** Validates the interpretation by re-checking against the original image.
4. **Multi-Round Refinement:** Iteratively refines the interpretation until it consistently passes verification.
5. **Consecutive Verification:** Accepts an interpretation only after it passes the required number of consecutive verifications.

---

[2]https://www.nano-banana.ai/
[3]https://www.doubao.com/chat/

## B.2 COEVOLUTIONARY ITERATION STRATEGY

The PHYSICSMINIONS employs a multi-round coevolutionary iteration strategy:

1. **Initial Solution:** The Solver generates an initial solution, which is improved by the Introspector to correct errors.

2. **Dual-Stage Verification:**
   - **Stage 1:** The Physics-Verifier checks domain-specific physics consistency (e.g., units, constants, assumptions).
   - **Stage 2:** The General-Verifier checks detailed logical and computational correctness.

3. **Iterative Refinement:**
   - **Failure:** If either Physics-Verifier or General-Verifier fails, a bug report is generated, corrected by the Introspector, and re-verified.
   - **Success:** If both pass, the count of consecutive successes increases; once the threshold is met, the solution is accepted, otherwise verification repeats.

4. **Consecutive Verification:** A solution is accepted only after meeting the predefined threshold of consecutive successful verifications. If the threshold of consecutive failures is reached, the process restarts with a new initial solution.

## B.3 IMPLEMENTATION DETAILS

---

**Algorithm 1** PhysicsMinions: A Coevolutionary Multimodal Multi-Agent System

---

**Input:** Problem instance (incl. images); Consecutive Verification threshold CV (default: 2)
**Output:** Final solution
1: $I \leftarrow$ VisualExtract(Problem)  ▷ Visual Studio: extract structured visual info
2: $S \leftarrow$ GenerateInitialSolution(Problem, $I$)  ▷ Solver uses structured input $I$
3: $c \leftarrow 0; f \leftarrow 0$  ▷ $c$=consecutive successes, $f$=consecutive failures
4: **while** not converged **do**  ▷ Loop at most 5 iterations
5:     $S \leftarrow$ IntrospectorImprove($S$)
6:     Pass or Fail, Bug_report $\leftarrow$ PhysicsVerify($S$)  ▷ Stage 1: Physics-Verifier
7:     **if** Fail **then**
8:         $S \leftarrow$ IntrospectorImprove(Bug_report)
9:         $c \leftarrow 0; f \leftarrow f + 1$  ▷ reset success; increment failures
10:         **if** $f \geq$ CV **then**
11:             $S \leftarrow$ GenerateInitialSolution(Problem, $I$)
12:             $c \leftarrow 0; f \leftarrow 0$
13:         **end if**
14:         **continue**
15:     **end if**
16:     Pass or Fail, Bug_report $\leftarrow$ GeneralVerify($S$)  ▷ Stage 2: General-Verifier
17:     **if** Fail **then**
18:         $S \leftarrow$ IntrospectorImprove(Bug_report)
19:         $c \leftarrow 0; f \leftarrow f + 1$
20:         **if** $f \geq$ CV **then**
21:             $S \leftarrow$ GenerateInitialSolution(Problem, $I$)
22:             $c \leftarrow 0; f \leftarrow 0$
23:         **end if**
24:         **continue**
25:     **else**
26:         $c \leftarrow c + 1; f \leftarrow 0$
27:         **if** $c \geq$ CV **then**
28:             **return** $S$
29:         **end if**
30:     **end if**
31: **end while**

## C    DETAILED PROMPTS OF PHYSICSMINIONS

**Prompt Design.** We design domain-specific prompts tailored for physics Olympiads, where both multimodal understanding and physics reasoning are critical. The **Visual Studio prompts** are crafted to extract key information from typical physics figures, including axes, units, data trends, and labeled variables in line plots or schematic diagrams. The **Physics-Verifier prompts** guide systematic checks from coarse-grained validations (e.g., unit consistency and physical constants) to fine-grained reasoning involving implicit assumptions and physical laws. These carefully constructed prompts support reliable error detection and correction across the coevolutionary pipeline. All prompts were initially designed by humans and lightly polished by GPT-5.

### C.1    PROMPTS OF VISUAL STUDIO

- **Inspector:** extracts initial structured information from the image.
- **Introspector (Image):** improves and refines the structured information.
- **Verifier (Image):** validates the extracted information against the original image.

```
Inspector

# Your Task
Analyze physics problem's **reference figures** (schematic,
    plot, free-body diagram, circuit diagram, optical diagram,
    waveform, table, object image, or combination). Extract
    **every possible visual detail** without fabrication,
    producing a **structured JSON summary** that would allow
    accurate reconstruction of the figure.

## Step 1 -Coarse Scan (Global Understanding)
For each figure:
1. Identify the **overall purpose** and **theme** of the figure.
2. Locate and read any **title, caption, or source**.
3. Count and identify **subfigures** (a), (b), (c)...
- For each, note:
    - **Theme** (main concept)
    - **Figure type** (choose from: 'plot', 'free_body',
        'circuit', 'optics', 'waveform', 'table', 'schematic',
        'object_image', 'other')

### Classification Priority Rules
- If a figure clearly matches **circuit diagram**, **optical
    setup/diagram**, or **free-body diagram**,
->**must** classify it as 'circuit', 'optics', or 'free_body'
    respectively,
->**do NOT** lump these into 'schematic'.
- Use 'schematic' **only for general physical setups / abstract
    line drawings** that do not fit into the above categories.
- Use 'object_image' if it is a **real-world photo or realistic
    rendering** of experimental apparatus or objects.
- If the figure does not fit any known category, classify as
    'other' and provide a short explanation.

## Step 2 -Detailed Scan (Type-Specific Rules)
For **each** figure/subfigure, follow **type-specific
    extraction rules**:

### A. Axis-Based Figures ('figure_type = "plot"')
Extract enough information to rebuild the plot.

- **x_axis / y_axis**:
    - 'label': From axis text (or '"unknown"' if missing)
```

```
  - `unit`: Prefer SI units (or `"unknown"`)
  - `range`: `[min, max]` estimated from labeled tick marks
  - `scale`: `"linear"` or `"log"`
  - `ticks`: List **all labeled major tick values** as numbers or
      strings exactly as shown. If a label is not numeric, keep
      its string.

  #### Curves --REQUIRED `curves[*].data` FORMAT UPDATED
  For each curve, extract using the schema below. **Do not
      fabricate** values; if a y-value cannot be read, set it to
      `"unknown"` but still include the x tick.

  - `name`: Legend label or `"unknown"`
  - `color`: Visible curve color (e.g., `"blue"`, `"red"`, or hex
      if distinctive)
  - `line_style`: `"solid"`, `"dashed"`, `"dotted"`, etc.
  - `overall_trend`: A 1-3 sentence summary of the curve's
      behavior over the full x-range (e.g., increasing then
      plateau; single peak near x≈2; S-shaped with inflection).
  - `data`:
  - `by_x_ticks`: **Must include**:
    1. **All visible labeled major x ticks**.
    2. **Starting point** (leftmost visible data).
    3. **Ending point** (rightmost visible data).
    Each entry format:
    ```json
    { "x": <value>, "y": <estimated_or_"unknown">,
        "read_method": "from_graph", "note": "<optional short
        note>" }
    ```
  - `special_points`: **Must include all identifiable notable
      points**, e.g.:
    ```json
    { "type": "peak" | "valley" | "inflection" | "x_intercept" |
        "y_intercept" | "endpoint_feature" | "intersection",
    "x": <value_or_"unknown">, "y": <value_or_"unknown">,
        "with_curve": "<name if intersection>",
    "description": "<why it's special>", "confidence":
        "high"|"medium"|"low" }
    ```

  #### Scatter Points
  For each scatter series, extract using the schema below. **Do
      not fabricate** values; if a y-value cannot be read, set it
      to `"unknown"`.

  - `name`: Series label or `"unknown"`
  - `color`: Marker color
  - `shape`: Marker shape (circle, square, etc.)
  - `overall_distribution`: 1-3 sentence description of how the
      points are arranged (clustered, linear trend, random spread,
      etc.)
  - `data`:
  - `by_x_ticks`: **Must include**:
    1. **All visible labeled major x ticks** (0, 1, 2...).
    2. **Uniformly sampled intermediate points** (pick
        representative points in between ticks if present).
    3. **Starting point cluster** (leftmost x with points).
    4. **Ending point cluster** (rightmost x with points).
    Each entry format:
    ```json
```

```
      { "x": <tick_value>, "y_values": [<all points near this tick
          or "unknown">], "read_method": "from_graph", "note":
          "<optional>" }
      ```
  - `special_points`: **Must include all identifiable notable
      points**, such as outliers, cluster centers, gaps, or
      notable trend points:
      ```json
      { "type": "outlier" | "cluster_center" | "gap" |
          "trend_point",
      "x": <value_or_"unknown">, "y": <value_or_"unknown">,
      "description": "<why it's special>", "confidence":
          "high"|"medium"|"low" }
      ```

  ### B. Free-Body Diagrams (`free_body`)
  - List the object(s)
  - All forces acting (gravity, normal, friction, tension,
      applied forces)
  - Force directions and decompositions
  - Equilibrium hints

  ### C. Circuit Diagrams (`circuit`)
  - All components with values and symbols
  - Polarities and connections
  - Switch states, measurement points

  ### D. Optical Diagrams (`optics`)
  - Media (air, glass, water, etc.)
  - Rays (incident, reflected, refracted)
  - Lens/mirror types and focal lengths
  - Object/image positions

  ### E. Waveforms (`waveform`)
  - Amplitude, period, frequency, phase
  - Envelope shapes, modulation patterns
  - Time base scale

  ### F. Tables (`table`)
  - Column names and units
  - All values
  - Missing-data markers

  ### G. Schematics / Physical Setups (`schematic`)
  1) **Elements to Extract (must be complete but concise)**
  - **Points**: list all labeled points (every symbol shown).
  - **Vectors**: include all vectors with start->end notation
      (e.g., `S->Sv`), preserve arrow style/color if meaningful.
  - **Segments**: use `start-end` format (e.g., `S-E`, `s-S`). If
      segment length/value is known, include it as `"value"`.
  - **Angles**: include only angles with labels (e.g., $\beta$) or
      known values (e.g., right angles). Label them as `angleABC`
      (e.g., `angleCET`).
  - **Arcs**: include arcs with center and through point (e.g.,
      `"center": "C", "through": "E"`), specify dashed/solid if
      relevant.
  - **Styles**: only if meaningful (color, dashed, arrow).

  2) **Output Format (JSON-like)**
  - **Labeling rule**:
  - Use the exact label from the diagram if shown.
  - If none, use `start-end` for segments and `start->end` for
      vectors.
```

```
       - For angles, always use `angleABC` form, where B is the vertex.

       ### H. Object Images (`object_image`)
       - Objects present
       - Shape, composition, materials
       - Spatial relationships and distances

       ---
       # Step 3 Output Rules
       - **If single figure**:
       ```json
       {
       "title": "",
       "figure_type": "",
       "x_axis": {...},
       "y_axis": {...},
       "curves": [...],
       "scatter_points": [...]
       }
       ```
       - **If multiple figures**:
       {
       "title": "",
       "figure": [
          {
          "id": "a",
          "theme": "",
          "figure_type": "",
          "x_axis": {...},
          "y_axis": {...},
          "curves": [...],
          "scatter_points": [...]
          },
          {
          "id": "b",
          ...
          }
       ]
       }
```

**Introspector (Image)**

```
    You are an IMAGE-READING EXTRACTOR for physics-related figures
        (plots, circuit diagrams, schematics, object photos).

    Goal:
    Read the IMAGE ONLY and output a FULL JSON in the EXACT SAME
        SCHEMA required by read_image_prompt.
    Your output must be self-contained, consistent, and reflect the
        visible content of the image.

    Critical rules:
    - Follow EXACTLY the JSON schema used by read_image_prompt
        (keys, nesting, arrays).
    - Use the IMAGE ONLY. Do not fabricate elements that are not
        visible.
    - If something is unclear/occluded, include it with lower
        confidence and describe uncertainty briefly.
    - Keep the same JSON structure:
    {
       "title": <str>,
```

```
    "figure": [
    { ... sub-figure #1 ... },
    { ... sub-figure #2 ... }
    ]
  }
- For each sub-figure, set an appropriate figure_type: "plot" |
    "schematic" | "circuit" | "photo" (or other valid types
    defined in read_image_prompt).

Main detection & extraction focus:
1. **Theme**: Ensure the high-level description matches the
    figure (e.g., is it a plot, circuit, geometry, or photo).
2. **Elements**: Identify and record points, vectors, segments,
    angles, arcs, circuit components, arrows, right-angle marks,
    hats/circles/subscripts/superscripts, geometric constraints
    (perpendicular/parallel).
3. **Data**:
- For plots: re-read axis labels, units, ranges, scales, ticks.
- Capture curves with name, color, line_style, overall_trend,
    compact data samples (e.g., by_x_ticks).
- Identify special_points (peaks, valleys, intercepts) with
    accurate values and confidence.
- Ensure numbers and symbols are correct and consistent.
4. **Labels**: Check that Greek letters, subscripts,
    superscripts, and vector/matrix notation match the figure
    text exactly.

Consistency & quality:
- Cross-check that labels and values remain consistent across
    all sections.
- Use numeric precision appropriate to image resolution; if
    approximate, indicate so with "confidence".
- If a figure type or field is not present in the image, omit
    it rather than hallucinating.
```

### Verifier (Image)

```
### Image Verification Task (Physics Figures)

**Role:**
You are an image-reading verifier for physics-related figures
    (plots, circuit diagrams, schematics, object photos).
Your task is to compare the provided **IMAGE ONLY** against the
    provided **INFORMATION** text that claims to describe the
    image's content.

---

### Verification Focus
1. **Theme/Context**
- Check whether the overall theme of the figure matches the
    claimed description (e.g., if it is a circuit vs. a
    mechanics setup).

2. **Elements**
- Verify presence, shape, connectivity, and symbols (e.g.,
    circuit elements, arrows, objects, geometry).

3. **Data**
- Verify axis labels, units, ranges, tick marks, intercepts,
    and curve trends.
```

```
     - If numerical values are claimed, check approximate readings
       from the plot against the description.
     - **Error tolerance rule:** If deviations are smaller than 1/20
       of the smallest axis tick spacing**, treat them as
       acceptable.

     4. **Labels and Symbols**
     - Verify textual labels, math symbols (Greek letters,
       subscripts, superscripts, hats, arrows, circles), and check
       consistency with the description.

     ---

     ### Error Types
     - **Error**: Wrong values, units, or labels.
     - **Omission**: Missing elements in either the figure or the
       description.
     - **Inconsistency**: Mismatch between what the image shows and
       what is described.
     - **Misleading**: Description misrepresents trends, geometry,
       or relationships.
     - **Tolerance Check**: If numerical differences exceed the 1/10
       smallest-tick rule, classify as an error.

     ---

     ### Output Format
     1. **Line 1:** `IF CORRECT: yes` or `IF CORRECT: no`
     - Write `yes` if any error exists (even a single one).

     2. **If IF CORRECT = no:**
     Add a section titled exactly: "Detailed Verification"
     Then, list bullet points for each problem with this structure:
     - **Category**: (theme | elements | data | labels | tolerance)
     - **Evidence**: (what is visible in the image and where)
     - **Mismatch**: (what the INFORMATION claims)
     - **Why**: (explanation of the discrepancy)
     - **Confidence**: (high | medium | low)

     3. **If IF CORRECT = yes:**
     Briefly state which checks you performed and why the
       INFORMATION matches the IMAGE.
```

## C.2 PROMPTS OF LOGIC STUDIO

- **Solver:** generates the initial solution, using a prompt adapted from Huang & Yang (2025).

- **Introspector (Self-Improve):** performs the first improvement of the initial solution.

- **Introspector (Self-Refine):** revises the solution based on the bug report provided by the Review Studio.

```
Solver

  ### Core Instructions ###

  *  **Rigor is Paramount:** Your primary goal is to produce a
     complete and rigorously justified solution. Every step in
     your solution must be logically sound and clearly explained.
     A correct final answer derived from flawed or incomplete
     reasoning is considered a failure.
```

```
1134
1135        *   **Honesty About Completeness:** If you cannot find a
1136            complete solution, you must **not** guess or create a
               solution that appears correct but contains hidden flaws or
1137            reasoning errors. Instead, you should present only
1138            significant partial results that you can rigorously prove. A
1139            partial result is considered significant if it represents a
1140            substantial advancement toward a full solution. Examples
               include:
1141        *   Deriving a key physical law or principle.
1142        *   Fully resolving one or more cases within a logically
1143            sound physics-based analysis.
1144        *   Establishing a critical property of the physical system
1145            in the problem.
1146        *   For a physical system, determining constraints or
1147            boundary conditions without fully solving the dynamics.
           *   **Use TeX for All Physics Equations:** All physical
1148            variables, equations, and relations must be enclosed in TeX
1149            delimiters (e.g., `Let $v$ be the velocity of the object.`).
1150
1151        ### Output Format ###
1152
1153        Your response MUST be structured into the following sections,
               in this exact order.
1154
1155        **1. Summary**
1156
1157        Provide a concise overview of your findings. This section must
               contain two parts:
1158
1159        *   **a. Verdict:** State clearly whether you have found a
1160            complete solution or a partial solution.
1161            *   **For a complete solution:** State the final answer,
1162                e.g., "I have successfully solved the problem. The final
                    answer is..."
1163            *   **For a partial solution:** State the main rigorous
1164                conclusion(s) you were able to prove, e.g., "I have not
1165                found a complete solution, but I have rigorously proven
1166                that..."
           *   **b. Method Sketch:** Physical Modeling First, present a
1167            high-level, conceptual outline of your solution. This sketch
1168            should allow an expert to understand the logical flow of
1169            your argument without reading the full detail. It should
1170            include:
           *   A narrative of your overall strategy.
1171        *   The full and precise physical statements of any key
1172            principles or major intermediate results.
1173        *   If applicable, describe any key experimental setups or
1174            case analyses that form the backbone of your argument.
1175
1176        **2. Detailed Solution**
1177
1178        Present the full, step-by-step physical derivation or analysis.
               Each step must be logically justified and clearly explained.
1179            The level of detail should be sufficient for an expert to
1180            verify the correctness of your reasoning without needing to
1181            fill in any gaps. This section must contain ONLY the
1182            complete, rigorous derivation or analysis, free of any
               internal commentary, alternative approaches, or failed
1183            attempts.
1184
1185        ### Self-Correction Instruction ###
1186
1187
```

```
Before finalizing your output, carefully review your "Method
    Sketch" and "Detailed Solution" to ensure they are clean,
    rigorous, and strictly adhere to all instructions provided
    above. Verify that every statement contributes directly to
    the final, coherent mathematical argument.
```

**Introspector (Self-Improve)**

```
You have an opportunity to improve your solution. Please review
    your solution carefully. Correct coarse- or fine-grained
    errors if any. Your second round of output should strictly
    follow the instructions in the system prompt.
*   **a. Equation Derivation:** Derive all necessary equations
    step-by-step, ensuring each step is mathematically rigorous
    and physically meaningful. Clearly define all symbols and
    variables used in the equations.
*   **b. Numerical Computation:** Perform any numerical
    calculations required to obtain the final answer or
    intermediate results, ensuring consistency with the derived
    equations. Specify all numerical values, including their
    units.
*   **Notes:**
    *   **Unit Conversion:** Ensure all units are consistent
        throughout the derivation and calculations. Clearly state
        any unit conversions performed.
    *   **Symbol Definitions:** Define all symbols and variables
        clearly at their first use, and maintain consistency in
        their usage throughout the solution.
```

**Introspector (Self-Refine)**

```
Here is the bug report. If you agree with certain items in it,
    please refine your solution to make it more complete and
    rigorous. Keep in mind that the evaluator who generated the
    report may have misunderstood your solution and introduced
    mistakes. If you disagree with certain items, add detailed
    explanations to clarify your reasoning and prevent such
    misunderstandings. Your revised solution should strictly
    follow the instructions provided in the system prompt.
```

## C.3 PROMPTS OF REVIEW STUDIO

- **Physics-Verifier:** checks domain-specific physics consistency, such as units, constants, and assumptions.
- **General-Verifier:** detects logical, reasoning, and computational errors through step-by-step analysis.

**Physics-Verifier**

```
You are a physics-specific verifier. Your sole task is to
    quickly screen for basic physics hygiene issues.
Do NOT attempt to re-solve or correct the solution. Only detect
    problems and report them.

**1. Core Instructions**
- Do NOT re-solve the problem and do NOT correct the solution.
```

```
- If something is merely omitted but not required, do not
  penalize.

**2. Evaluation Pipeline**

**A) COARSE CHECKS (fast hygiene)**
Check ONLY for obvious mismatches between Problem and Solution:
- **C1. Units & Conversions (sanity)**
  - Spot incorrect conversions among common pairs: cm↔m, g↔kg,
    eV↔J, °C↔K, min↔s, h↔s.
- **C2. Constants & Given Values**
  - If the stem requires a specific value (e.g., g=9.8 or g=10),
    verify the solution uses it.
  - Common constants sanity check (e.g., k_B, N_A, e, c, h, ℏ).
    Allow standard rounding but flag clearly wrong magnitudes
    or inconsistent usage.

Output a **coarse verdict** BEFORE moving to fine checks.

**B) FINE CHECKS (physics consistency)**
Check ONLY for obvious errors while still NOT solving from
  scratch:
- **F1. Assumptions vs Stem**
  - Flag any unstated/extra assumptions not supported by the
    stem.
  - Standard harmless assumptions (e.g., ideal string, no
    friction if not mentioned) are acceptable.
- **F2. Physical Consistency**
  - Verify that physical quantities are used in the correct
    context (e.g., force vs energy, velocity vs acceleration).
  - If intermediate steps are skipped but the relation is
    physically sound, accept it.
  - Flag cases where a formula is applied to the wrong type of
    quantity, or where the result has incorrect units.

**3. Output Format**
Provide:
- Final Verdict: exactly one sentence: "PASS" if no material
  issue for the above checks, otherwise "FAIL".
- Findings: bullet list of every issue found (quote the exact
  offending text/equation).
- If FAIL: include a short, consolidated Bug Report (just the
  issues; do not fix them).
```

### General-Verifier

```
You are a DOMAIN-AGNOSTIC REASONING VERIFIER. Your job is to
  evaluate a proposed solution at TWO levels--COARSE then
  FINE--and produce a rigorous, structured verification. You
  are a verifier, NOT a solver.

**1. Core Rules**
- Your sole task is to find and report all issues in the
  provided solution. Do NOT correct, extend, or re-solve.
- You must perform a **step-by-step** check of the entire
  solution. This analysis will be presented in a **Detailed
  Verification Log**.
- If a step is missing but not required by the task, do not
  penalize.
- If a step is wrong and later steps depend on it, mark
  downstream as "tainted by prior error" and skip their
```

```
        detailed checking, but still scan for independent
        branches/cases.

    **2. Evaluation Pipeline**

    **A) COARSE CHECKS (fast hygiene)**
    Check ONLY for obvious mismatches between Problem and Solution:
    - **C1. Question-Answer Match**
      - If the problem asks for a numerical value but the solution
        gives only a formula, or vice versa.
      - Missing required sub-answers for multi-part questions.
      - Required symbol naming not followed (e.g., the problem
        specifies that the answer must be expressed in terms of
        $\alpha$ and $\beta$, but the solution introduces
        different symbols or omits them).
      - Stated unit in the problem vs final unit in the answer
        mismatch.
      - Significant-figure policy violated (e.g., asked for 3
        significant figures but final answer not in 3 significant
        figures).
    - **C2. Completeness of Response**
      - Solution ends abruptly, leaving an essential part clearly
        unfinished.
      - Skipped an entire subquestion that is explicitly required.
    - **C3. Problem Interpretation**
      - Misread or misinterpreted what the problem is asking (e.g.,
        solves for maximum instead of minimum).

    Output a **coarse verdict** BEFORE moving to fine checks.

    **B) FINE CHECKS (step-by-step reasoning consistency)**
    You must perform a **step-by-step** check of the entire
        solution without solving from scratch:
    - **F1. Logical Consistency**
      - Detect invalid inferences (e.g., claiming A>B and C>D
        implies A-C>B-D).
      - Identify contradictions within the argument.
    - **F2. Reasoning Flow**
      - Check that the solution applies definitions, theorems, and
        principles correctly.
      - Normal algebraic or logical steps may be skipped if they are
        standard and the result is consistent.
      - Only flag when a **critical step is missing** (e.g., no
        derivation provided for a non-trivial result), or when the
        solution **jumps directly to an answer without reasoning**.
      - If the reasoning is concise but valid, consider it
        acceptable.
    - **F3. Calculations & Algebra**
      - Arithmetic errors (e.g., 2+3=6).
      - Symbolic manipulation mistakes.

    **3. Output Format**
    Your response MUST be structured into two main sections: a
        **Summary** followed by the **Detailed Verification Log**.

    *   **a. Summary**
        This section MUST be at the very beginning of your response.
            It must contain two components:
        *   **Final Verdict**: A single, clear sentence declaring the
            overall validity of the solution. For example: "The
            solution is correct," "The solution fails due to major
            reasoning errors," or "The solution is partially valid
            but incomplete."
```

```
        *   **List of Findings**: A bulleted list that summarizes
            **every** issue you discovered. For each finding, you
            must provide:
            *   **Location:** A direct quote of the key phrase or
                equation where the issue occurs.
            *   **Stage:** Whether the issue belongs to **COARSE**
                (Q-A mismatch, completeness, interpretation) or
                **FINE** (logic, reasoning, calculation).
            *   **Issue:** A brief description of the problem.

    *   **b. Detailed Verification Log**
        Following the summary, provide the full, step-by-step
            verification log as defined in the Core Instructions.
            When you refer to a specific part of the solution,
            **quote the relevant text** to make your reference clear
            before providing your detailed analysis of that part.

    *   **Example of the Required Summary Format**
    *This is a generic example to illustrate the required format.
        Your findings must be based on the actual solution provided
        below.*

    **Final Verdict:** The solution is **invalid** because it
        contains major reasoning errors.

    **List of Findings:**
    *   **Location:** "The maximum value is found by setting
        $f'(x)=0$ and solving $x=2$"
        *   **Stage:** COARSE -Problem Interpretation
        *   **Issue:** The problem explicitly asks for the
            **minimum**, but the solution computes the maximum.

    *   **Location:** "Since $A > B$ and $C > D$, it follows $A - C
        > B - D$"
        *   **Stage:** FINE -Logical Consistency
        *   **Issue:** Invalid inference; inequalities cannot be
            combined this way.

    *   **Location:** "Therefore, $2 + 3 = 6$"
        *   **Stage:** FINE -Calculation
        *   **Issue:** Arithmetic error.
```

# D ILLUSTRATION OF COMPARISON FRAMEWORKS

## D.1 BEST-OF-N STRATEGY

To compare with the upper-bound performance of a single model, we adopt a simple yet effective
**Best-of-N** strategy (Stiennon et al., 2020). For each physics exam, the model is prompted to solve the
same paper **three times independently** using stochastic sampling. Among the three completions,
we select the one with the highest overall exam score as the final output. This reflects the common
practice of sampling multiple completions and choosing the best-performing one, thereby assessing
the model's potential when given multiple attempts.

All completions are generated with temperature $0.6$ and the max_tokens parameter set to the
maximum allowed by each model to prevent premature truncation.

## D.2 SELF-MOA

We further compare with the **Self-Mixture-of-Agents (Self-MoA)** framework (Li et al., 2025). Un-
like conventional mixture-of-agents methods that ensemble outputs from multiple models, Self-

MoA operates entirely within a single model. Its core idea is to leverage the diversity induced by repeated stochastic decoding, then let the model act as both critic and aggregator.

For each exam question, we first collect two independent completions. These candidates often display complementary strengths, such as partial correctness, alternative reasoning paths, or differences in detail. The model is then prompted in two stages: (i) it critiques both candidates, analyzing merits and flaws; and (ii) it synthesizes a refined solution that preserves strengths while discarding errors. This self-reflective debate-and-refinement process enhances coherence and informativeness without requiring multiple models or external agents.

In our experiments, we follow the standard setup with temperature fixed at $0.6$ to encourage moderate diversity, and `max_tokens` set to the maximum allowable length.

### D.3 SELF-REFINE

We also evaluate **Self-Refine** (Madaan et al., 2023), an iterative approach that improves model outputs through cycles of self-reflection and refinement. The model first generates an initial solution, then critiques it by identifying potential issues such as incompleteness, inaccuracies, lack of clarity, or logical flaws.

Based on this critique, the model revises its solution while preserving its overall structure and intent. This process repeats for a fixed number of iterations (set to three in our experiments), enabling progressive improvement. Viewed as a self-supervised optimization loop, Self-Refine is particularly effective for multi-step reasoning tasks, where iterative clarification and refinement are crucial.

We use the standard configuration with temperature set to $0.6$ and `max_tokens` set to the model's maximum limit.

### D.4 ADVANTAGES OVER COMPARISON FRAMEWORKS

As discussed in Section 4.3 of the main text, our PHYSICSMINIONS framework achieves both *consistent improvement* and *medal progression*, which are not attainable by existing baselines. We highlight the advantages over the three comparison frameworks as follows:

- **Best-of-N:** Although this strategy selects the highest-scoring completion among $N$ attempts, it essentially remains a single-model approach, bounded by the performance ceiling of individual reasoning runs. In contrast, by incorporating verification–reflection cycles, PHYSICSMINIONS fully leverages the model's self-correction ability and substantially enhances reasoning performance. For example, on the latest IPhO, PHYSICSMINIONS with Intern-S1 achieves higher scores on every problem compared with the single-model baseline (see Fig. 3).
- **Self-MoA:** Compared to single-model inference, Self-MoA synthesizes multiple candidate solutions into a new response. However, its performance is constrained by the quality of the candidates and the model's aggregation ability. When self-consistency among candidates is low, Self-MoA may merge spurious reasoning paths, introducing additional errors. Without the iterative reflection of PHYSICSMINIONS, its stability is limited, and in practice, it sometimes yields scores lower than those of the single model.
- **Self-Refine:** While Self-Refine incorporates self-correction through iterative refinement, it lacks two critical components of our framework: (i) *dual-stage verification*, where our Physics-Verifier enforces domain-specific checks (e.g., unit consistency and physical constants) and our General-Verifier conducts comprehensive evaluations of completeness, logic, reasoning, and calculations; and (ii) *coevolutionary collaboration*, where the Review Studio and Logic Studio iteratively validate and refine solutions toward correctness. By contrast, Self-Refine operates without domain knowledge and lacks collaborative coevolution, limiting its effectiveness on complex physics problems.

## E OLYMPIAD DESCRIPTION AND DETAILED RESULTS

### E.1 OVERVIEW OF PHYSICS OLYMPIADS

Our evaluation covers seven physics Olympiads spanning international and regional competitions:

- **IPhO (International Physics Olympiad):** The premier global physics competition for high school students, featuring both demanding theoretical exams and experimental challenges.
- **APhO (Asian Physics Olympiad):** A regional contest for students from Asia and Oceania, following the IPhO format with combined theory and laboratory components.
- **EuPhO (European Physics Olympiad):** A continental competition for European students, emphasizing creative and open-ended approaches to theoretical and experimental problems.
- **NBPhO (Nordic-Baltic Physics Olympiad):** A regional contest among Nordic and Baltic countries, focusing on theoretical problem solving with occasional experimental tasks.
- **PanPhO (Pan Pearl River Delta Physics Olympiad):** An invitational exam for top schools in China's Pearl River Delta and neighboring regions, covering a wide spectrum of physics topics.
- **PanMechanics (Pan Pearl River Delta Mechanics Contest):** A specialized subset of PanPhO dedicated exclusively to mechanics, typically structured as a shorter single-field exam.
- **F=MA:** A U.S. national mechanics contest organized by the American Association of Physics Teachers (AAPT), serving as the entry test for the U.S. Physics Olympiad (USAPhO).

All data are sourced from the HiPhO benchmark (Yu et al., 2025), which compiles original exam materials and official records from the respective official websites. This includes human contestants' scores, which serve as reference points. The gold, silver, and bronze thresholds in our evaluation also follow the HiPhO benchmark, derived directly from the official scores of human medalists.

### E.2 PERFORMANCE GAINS ACROSS MODALITY TYPES

The HiPhO benchmark (Yu et al., 2025) defines four modality types: Text-Only (TO), Text+Illustration Figure (TI), Text+Variable Figure (TV), and Text+Data Figure (TD). Fig. 3 shows the performance gains of Intern-S1 across these modalities. To evaluate overall performance, we follow HiPhO and use the mean normalized score (MNS), defined as

$$\mathbf{MNS}(M) = \frac{1}{N_M} \sum_{Q \in M} \frac{\text{Exam Score}(Q)}{\text{Full Mark}(Q)} \times 100\%,$$

where $M \in \{\text{TO}, \text{TI}, \text{TV}, \text{TD}\}$, $N_M$ is the number of questions in $M$, and $Q$ denotes a single question. Table 4 demonstrates that PHYSICSMINIONS achieves consistent gains across all modalities.

Table 4: Performance gains of mean normalized scores (%) across four modality types.

| Modality Type | TO | TI | TV | TD |
|---|---|---|---|---|
| Gemini-2.5-Flash-Thinking | 81 | 66 | 68 | 67 |
| **+ PHYSICSMINIONS** | 91 | 77 | 76 | 73 |
| Intern-S1 | 80 | 63 | 57 | 48 |
| **+ PHYSICSMINIONS** | 90 | 74 | 64 | 63 |
| InternVL3.5-241B-A28B | 69 | 49 | 45 | 47 |
| **+ PHYSICSMINIONS** | 88 | 72 | 63 | 62 |
| Qwen2.5VL-32B-Instruct | 58 | 36 | 37 | 34 |
| **+ PHYSICSMINIONS** | 70 | 44 | 41 | 41 |

### E.3 PERFORMANCE GAINS ACROSS PHYSICS FIELDS

Physics Olympiad problems cover five major fields: Mechanics, Electromagnetism, Thermodynamics, Optics, and Modern Physics. Table 5 reports the mean normalized scores for each field. Notably, PHYSICSMINIONS achieves substantial gains in Mechanics and Optics, where multimodal image inputs are common. These improvements highlight the dual advantage of accurate extraction by the Visual Studio and the coevolutionary iteration between the Logic Studio and Review Studio.

Table 5: Performance gains of mean normalized scores (%) across five major physics fields.

| Physics Field | Mechanics | Electromagnetism | Thermodynamics | Optics | Modern Physics |
|---|---|---|---|---|---|
| Gemini-2.5-Flash-Thinking | 69 | 65 | 91 | 43 | 84 |
| **+ PHYSICSMINIONS** | 81 | 68 | 91 | 49 | 98 |
| Intern-S1 | 64 | 64 | 81 | 39 | 64 |
| **+ PHYSICSMINIONS** | 75 | 64 | 91 | 51 | 79 |
| InternVL3.5-241B-A28B | 49 | 53 | 72 | 33 | 70 |
| **+ PHYSICSMINIONS** | 73 | 59 | 90 | 51 | 78 |
| Qwen2.5VL-32B-Instruct | 36 | 44 | 65 | 29 | 55 |
| **+ PHYSICSMINIONS** | 45 | 53 | 68 | 40 | 69 |

## F ILLUSTRATION OF DIFFERENT CHART ANALYSIS TOOLS

- **ChartGemma:** A chart reasoning model built on a strong vision–language backbone (Masry et al., 2024b). It takes chart images and text prompts as input, and outputs summaries, answers, or fact-checking results, enabling effective chart understanding.

- **ChartInstruct-FlanT5-XL:** Similar to ChartGemma, this method (Masry et al., 2024a) performs chart reasoning tasks such as summarization, question answering, and fact-checking, but differs by relying on instruction tuning.

- **Chart2Table:** A multimodal model developed by the PaddlePaddle team for chart parsing (PaddlePaddleTeam, 2025). It takes chart images as input and outputs structured tables, enabling automatic extraction of underlying data.

- **DescribePicture:** A prompt-based AI web tool for image description (AiDescribePicture, 2024). It takes an image (including charts) plus a text prompt as input, and outputs plain text or Markdown descriptions.

- **Flux-ai:** A web-based AI model that generates detailed descriptions from images (Flux AI, 2024). It allows users to input both an image and a text prompt, guiding the AI to provide specific descriptions or analyses.

- **Graph2Table:** An AI tool that automatically converts graph images into structured tabular data, supporting various graph types such as bar and line charts (Graph2Table, 2024). Users can upload images, which are then processed to generate downloadable CSV files.

- **Image Describer X:** An AI tool that generates detailed descriptions from images, supporting both image files and optional text prompts (Image Describer X, 2025).

- **PyVision:** It is an open-source framework (Zhao et al., 2025) and enhances visual reasoning in multimodal language models by enabling them to dynamically generate and execute Python tools for visual tasks. It supports iterative problem solving and adapts strategies during task execution, as shown in Fig. 8a.

- **TextIn:** A PDF-to-Markdown tool (TextIn, 2025) that converts PDF documents, including charts, into structured Markdown format, as shown in Fig. 8b.

- **WebPlotDigitizer:** A tool (Automeris, Inc., 2024) that enables users to extract numerical data from images of various types of charts, such as XY plots, bar charts, polar plots, and ternary diagrams. The process involves manual calibration of the chart axes, followed by user selection of data points to extract numerical values, as shown in Fig. 8c.

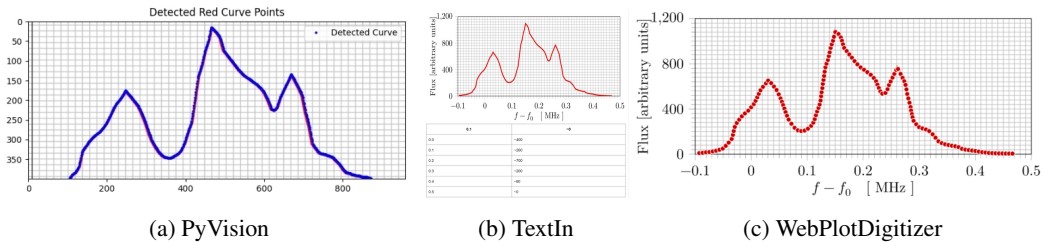

|   (a) PyVision   |   (b) TextIn   |   (c) WebPlotDigitizer   |

Figure 8: Results produced by different chart analysis tools.

**Limitation of Chart Analysis Tools.** The chart analysis tools discussed above can be grouped into three categories. The first includes models such as Flux-ai and PyVision that focus on image description, but these often struggle with charts containing complex or fine-grained information. The second category, represented by tools like Chart2Table and Graph2Table, converts charts into structured tables, yet still faces notable challenges in achieving high precision. The third category includes traditional tools such as WebPlotDigitizer, which can be highly accurate but relies on labor-intensive manual calibration, limiting efficiency at scale. Overall, while existing tools provide useful functionalities, they all exhibit clear limitations, and future work will explore the development of new tools tailored to chart analysis in multimodal reasoning tasks.But the verifier can recognize that the reading results do not match the graph, thus identifying the existing deviation. After multiple rounds of verification and optimization, we can reduce the deviation.

## G   EFFECTIVENESS OF PHYSICSMINIONS IN CHART INTERPRETATION

To evaluate the effectiveness of PHYSICSMINIONS in chart interpretation, we perform chart-reading tasks on multiple Olympiad problems using Gemini-2.5-Flash-Thinking as the base model. As illustrated in Fig. 9, problem IPhO_2025_1_C_4 requires extracting the coordinates of several peak points from a plotted curve. When solved by a single model alone, the results exhibit noticeable bias, primarily due to its one-shot reasoning and limited ability to accurately locate key points.

In contrast, our *image-verifier* in Visual Studio is able to re-parse the chart, detect that the preliminary readings deviate from ground truth, and generate a bug report. A similar pattern is observed in problem NBPhO_2025_2_1, where the verifier successfully identifies inconsistencies between the single-model output and the actual chart data. Based on these verification results, our *image-introspector* then revises the visual information by reanalyzing the chart and adjusting the extracted values. Through multiple reflection-verification cycles, PHYSICSMINIONS progressively corrects visual extraction errors and significantly reduces information bias.

These results collectively demonstrate that PHYSICSMINIONS achieves more accurate and reliable chart interpretation than a single model operating without verification or reflection.

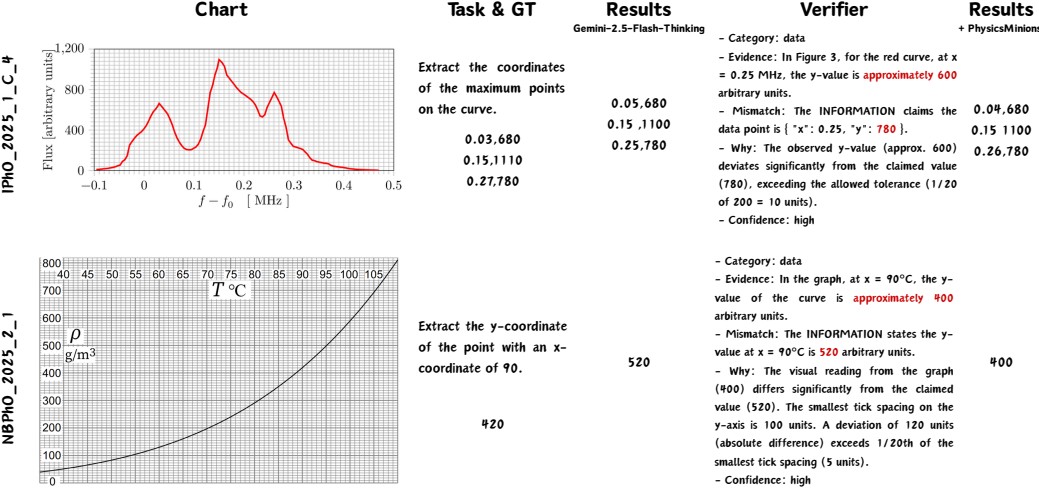

Figure 9: Effectiveness of PHYSICSMINIONS in chart interpretation compared to a single model. The reflection-verification cycle improves visual information extraction. GT denotes ground truth.

