# OpenReview forum: "PhysicsMinions: Winning Gold Medals in the Latest Physics Olympiads with a Coevolutionary Multimodal Multi-Agent System"
_ICLR.cc/2026/Conference — Submitted to ICLR 2026_

### Official Review · Reviewer_9HJu · 2025-10-30

**Soundness:** 3
**Presentation:** 3
**Contribution:** 2
**Rating:** 4
**Confidence:** 3

**Summary:**

This work introduces a multi-agent framework for solving olympic-level physics problems. The proposed framework PhysicsMinions consists of a visual studio that converts the images into structured representation, a logic studio that provides the analysis and solution to the problem, and a review studio that check the solution from the perspective of physics-related format and correctness of the solution. The interplay happens between the logic and review module to support refinement. Results show that such method achieves the gold medal level performance on all olympic-level physics exam benchmarks.

**Strengths:**

- Gold medal performance with a light multi-agent framework is exciting.
- The comparison of different visual representation format reveals the preference of existing model in processing the reasoning information figures and images.

**Weaknesses:**

- The key feature of such a multi-modal multi-agent system for challenging physics reasoning problems would be the ability to interpret the scientific figures. The information and findings regarding this is limited. Besides, the interpretation is fixed during the iterative process (perhaps related to the benchmark features).
- The comparison between physicsminions and best-of-N is not well elaborated. Are they restricted with the same number of solution attempts or similar number of tokens? Specially considering that Best-of-N and self-MoA achieves on-par performance with PhysicsMinion in EuPhO and PanMechanics.

**Questions:**

See above.

---

> ### Author Response · Authors · 2025-11-22
>
> Thank you for your valuable comments. We appreciate your recognition of our gold medal results, lightweight multi-agent design, and the insights from our analysis of visual formats. We will update the paper accordingly.
>
> ---
>
> **Comment 1.** (1) The key feature of such a multi-modal multi-agent system for challenging physics reasoning problems would be the ability to interpret the scientific figures. The information and findings regarding this is limited. (2) Besides, the interpretation is fixed during the iterative process (perhaps related to the benchmark features).
>
> **Response 1.** Thank you for highlighting this important point. **(1)** We agree that scientific figure interpretation is a key challenge in multimodal physics reasoning. As discussed in **Section 5.2**, we conducted a case study and quantitative analysis showing that current single-model MLLMs and most existing chart analysis tools struggle to produce reliable results, especially for data figures. To address this, we developed the **reflection-verification enhanced image-reading system (Visual Studio)**, which significantly improves visual reasoning performance. While our method still has limitations, it consistently enhances the model's visual capabilities, as demonstrated by the results and examples provided in **Section 5.2** and **Appendix G**. **(2)** Although figure interpretation is initialized once per problem, it is further refined through reflection-verification cycles within the Visual Studio, enabling iterative improvement.
>
> In addition to the reflection-verification enhanced image-reading system (Visual Studio), we introduce a general-specialized integrated dual-verifier system (Review Studio) and develop a **coevolutionary multi-modal multi-agent system** tailored for physics Olympiads that **achieves the first open-source gold medal on IPhO 2025** and **generalizes across seven latest physics Olympiads** for the first time. We hope the reviewers could comprehensively recognize our contributions, which include three key aspects: **a novel and significant but underexplored field**, **system-level design innovations**, and **performance breakthroughs**.
>
> ---
>
> **Comment 2.** The comparison between physicsminions and best-of-N is not well elaborated. Are they restricted with the same number of solution attempts or similar number of tokens? Specially considering that Best-of-N and self-MoA achieves on-par performance with PhysicsMinion in EuPhO and PanMechanics.
>
> **Response 2.** Thank you for the question. In our experiments, all methods—including Best-of-N, self-MoA, and PhysicsMinions—were allowed the **same number of solution attempts** (three), ensuring a fair comparison. To further evaluate its potential, we increased N up to 32 and observed that performance on IPhO 2025 only modestly improved to 17.1, still significantly lower than PhysicsMinions (20.8). This suggests that simply sampling more outputs from a single model hits a ceiling due to its inherent limitations. While Best-of-N and self-MoA achieve scores close to PhysicsMinions on EuPhO and PanMechanics, we emphasize that **PhysicsMinions achieves stable and remarkable improvements (ranging from 12.2% to 30.8%) across all seven Olympiads, with a performance leap from 2 gold medals to 6**, whereas the other methods lack such consistency and gains. This highlights the advantage of our coevolutionary multi-agent system in producing robust improvements across diverse and challenging tasks.

---

> ### Author Response · Authors · 2025-11-27
> **Follow-up on Reviewer Concerns**
>
> Dear Reviewer,
>
> Thank you once again for your valuable comments and for recognizing our gold medal performance. We would like to confirm whether our rebuttal has satisfactorily addressed your concerns or if there are any remaining points that may still require clarification. We remain ready to provide further explanation if needed.
>
> Thank you for your time and consideration.

---

### Official Review · Reviewer_KQyp · 2025-11-01

**Soundness:** 3
**Presentation:** 3
**Contribution:** 3
**Rating:** 6
**Confidence:** 3

**Summary:**

Existing single-models fail to solve Physics Olympiad tasks because it requires both complex reasoning and multimodal understanding. In this paper, the authors propose PHYSICSMINIONS, a framework of three connected studios: Visual, Logic, and Review, which refine solutions through repeated feedback. This coevolutionary process unites structured perception, reasoning, and verification, enabling the first open-source gold medal and near human-level results in the latest International Physics Olympiad.

**Strengths:**

1. This paper clearly identifies the limitations of the single model architecture in Physics Olympiad tasks (fail to handle both complex reasoning and multimodal understanding).
2. They evaluate on a real and challenging benchmark to ensure fair and representative evaluation and significantly improve the performance (i.e. approaching human-expert level).
3. The proposed system includes three components and delegate the tasks. This coevolutionary multi-agent system allows agents to cooperate, reflect, and verify each other's outputs, overcoming the limits of single model reasoning.

**Weaknesses:**

1. I think it would be good to report the computation cost since the coevolution loop likely adds some iterations for the communications. I want to see a comparison between the single model and multi agent system.
2. If I understand correctly, all agents use the same LLM, I think the system lacks heterogeneity and diversity of reasoning. LLM from different model family could have different aspect of reasoning, visualization abilities. What's the performance to combine different LLMs? I'm curious if this can further improve the results.
3. Related work section does not cover the research on MLLM-based chart and figure understanding. I think the authors could add an section to talk about the visualization ability of current LLM on understanding the chart and figure (for example, CharXiv https://arxiv.org/abs/2406.18521, an evaluation benchmark).

**Questions:**

In this paper, the authors mention that the dual-verifier design assumes that consistent verification implies correctness. I'm curious what mechanisms exist to detect or prevent "false consensus", where both verifiers reinforce the same error?

---

> ### Author Response · Authors · 2025-11-22
>
> Thank you for your valuable comments. We sincerely appreciate your recognition of our work's strengths, including the effectiveness of our multi-agent system, the clear identification of single-model limitations, and the use of a challenging benchmark. We will revise the paper accordingly.
>
> **Comment 1.** I think it would be good to report the computation cost since the coevolution loop likely adds some iterations for the communications. I want to see a comparison between the single model and multi agent system.
>
> **Response 1.** Thank you for the valuable suggestion. We agree that evaluating resource usage is important for real-world applicability. While our coevolutionary multi-agent system introduces additional inference overhead compared to single-model baselines, it offers a unique advantage: it consistently **surpasses the performance ceiling of a standalone model** through introspection-verification cycles, and is fully compatible with any backbone model, including smaller models like Qwen2.5VL-32B-Instruct, which improves from bronze-level (9.9) to silver-level (12.4) on IPhO 2025 when enhanced by our system.
>
> As suggested, we conducted a comparison under **equal resource consumption**. On IPhO 2025, the average per-problem runtime for Intern-S1 is 180 seconds with 10,227 tokens, while for our system, it is 500 seconds with 89,482 tokens. This corresponds to a runtime ratio of approximately **1:3** and a token usage ratio of about **1:9**. Despite this, our system achieves **avg@3 = 20.8**, whereas the best-of-N result reaches only **17.1** under equal runtime (N = 9) and equal token budget (N = 27), showing limited improvement. This demonstrates that increasing sampling alone cannot overcome the performance limit of single models, whereas our system consistently reaches ~20 points even with a single sample. **We have included these discussions in Section 5.5.**
>
> Moreover, our framework enables **weaker models to perform comparably to stronger ones**. For example, Intern-S1 improves from 2 gold medals to 6, matching Gemini-2.5-Flash-Thinking, offering a **more lightweight and deployable solution under limited computational resources and deployment budgets**. This makes our system especially valuable in scenarios where closed-source models are impractical, and its effectiveness has been validated across multiple open-source models.
>
> ---
>
> **Comment 2.** If I understand correctly, all agents use the same LLM, I think the system lacks heterogeneity and diversity of reasoning. LLM from different model family could have different aspect of reasoning, visualization abilities. What's the performance to combine different LLMs? I'm curious if this can further improve the results.
>
> **Response 2.** Thank you for the thoughtful comment. We would like to clarify that we use the same LLM across all modules to demonstrate *how a single model can benefit from our coevolutionary system design*, showcasing **model self-evolution without relying on external diversity**. Exploring heterogeneous combinations of LLMs would require a more extensive study to evaluate the specific strengths and synergies between different models. As a preliminary investigation, we tested a configuration where **Visual Studio uses Gemini-2.5-Flash-Thinking, Logic Studio uses InternVL3.5-241B-A28B, and Review Studio uses Intern-S1**. As shown in Table R1, this heterogeneous setup outperforms configurations based solely on InternVL3.5-241B-A28B or Intern-S1, benefiting from the Gemini's superior visual capabilities. Compared to homogeneous setups, this heterogeneous design allows **smaller models to achieve significant performance gains** and enables **larger models to reduce inference cost**, offering a more practical trade-off between accuracy and efficiency. This suggests that heterogeneous configurations can enhance reasoning performance while also improving accessibility for settings with limited computational resources or funding.
>
> **Table R1. Performance of PhysicsMinions when combined with different models.**
> |Model|IPhO|APhO|EuPhO|NBPhO|PanPhO|PanMechanics|F=MA|
> |-|-|-|-|-|-|-|-
> |Gemini-2.5-Flash-Thinking|21.5|28.0|16.5|33.3|57.8|72.0|24.0
> |Intern-S1|20.8|25.2|10.1|28.9|46.8|68.7|22.7
> |InternVL3.5-241B-A28B|20.9|24.6|9.8|29.6|46.2|66.7|21.0
> |*Heterogeneous models*|21.2|26.2|12.4|30.8|49.5|69.4|23.1

---

> ### Author Response · Authors · 2025-11-22
>
> **Comment 3.** Related work section does not cover the research on MLLM-based chart and figure understanding. I think the authors could add an section to talk about the visualization ability of current LLM on understanding the chart and figure (for example, CharXiv, an evaluation benchmark).
>
> **Response 3.** Thank you for the valuable suggestion. As recommended, we have added a new **Section 5.3** and cited **CharXiv** to discuss the visual understanding capabilities of current MLLMs. Based on our experiments, we found that existing MLLMs struggle with accurate chart and figure interpretation, which is consistent with the observations reported in CharXiv. This difficulty is particularly evident in **data figures**, where models often fail to reliably read axes, extract numerical values, and identify key data points, as shown in Figure 7.
>
> In **Section 5.2**, we also evaluated several existing chart analysis tools, but most failed to deliver reliable results. The most effective solution so far has been our **reflection-verification enhanced image-reading system (Visual Studio)**, which significantly improves visual reasoning performance. This effectiveness is supported by case studies presented in **Section 5.2** and **Appendix G**.
>
> ---
>
> **Comment 4.** In this paper, the authors mention that the dual-verifier design assumes that consistent verification implies correctness. I'm curious what mechanisms exist to detect or prevent "false consensus", where both verifiers reinforce the same error?
>
> **Response 4.** Thank you for the insightful question. The ability to detect or prevent "false consensus" depends on two key factors: the reasoning capability of the base model and the validation strength of the verifier. If the model itself lacks reasoning accuracy, even the most carefully designed verifier prompts may fail to detect errors. On the other hand, our ablation study in **Section 4.5** shows that the **dual-verifier design outperforms the single-verifier setup**, highlighting the value of diverse verification perspectives. However, further increasing the number of verifications does not lead to better performance, suggesting that simply repeating checks is not effective. These findings support the importance of thoughtful verifier design and motivate future exploration into more comprehensive and robust verification strategies.

---

> ### Author Response · Authors · 2025-11-27
> **Follow-up on Reviewer Concerns**
>
> Dear Reviewer,
>
> Thank you once again for your valuable comments and for acknowledging the strengths of our work. As recommended, we have conducted additional experiments and added a detailed related work section. We would like to confirm whether our rebuttal has satisfactorily addressed your concerns or if there are any remaining points that may still require clarification. We remain ready to provide further explanation if needed.
>
> Thank you for your time and consideration

---

### Official Review · Reviewer_YFh1 · 2025-11-01

**Soundness:** 3
**Presentation:** 4
**Contribution:** 2
**Rating:** 4
**Confidence:** 3

**Summary:**

This submission introduces PhysicsMinions, a coevolutionary multimodal multi-agent framework designed to elevate AI performance on challenging Olympiad-level physics problems. The system is structured into three specialized studios: a Visual Studio for diagram interpretation, a Logic Studio for solution generation and refinement, and a Review Studio for dual-stage verification. The agents interact through iterative feedback loops, combining structured visual extraction, explicit stepwise reasoning, and systematic error detection/correction. The framework is evaluated on the HiPhO benchmark (across 7 recent physics Olympiads), using four state-of-the-art multimodal language models. Results indicate consistent improvements for both open and closed-source models, including reaching gold medalist-level performance previously unattainable for open-source systems.

**Strengths:**

1. The manuscript is notably clear, demonstrating substantial effort in both visualization and written presentation.

2. The study employs rigorous evaluation metrics (gold-medal thresholds and Pass@k) across seven recent and distinct Physics Olympiad competitions, comparing open-source and closed-source models to establish consistent performance benefits. Tables 1 and 2 convincingly show that *PhysicsMinions* elevates multiple models beyond the gold-medal line and achieves a historic milestone for open-source benchmarks.

3. The ablation studies systematically examine the impact of the Visual Studio, the Review Studio (including physics-specific and general verification), and the key hyperparameter—Consecutive Verification (CV). The results support the claim that each architectural component is necessary to attain the highest performance tier.

4. Section 5.2 candidly acknowledges current shortcomings in precise visual data extraction, pinpointing specific tools and failure cases within the system or available toolkits (see Fig. 7). This openness establishes a solid and honest foundation for future methodological improvements.

5. The design rationale—encompassing visual, logical, and review components—is well-motivated, with clearly defined and differentiated roles for diagram parsing, symbolic reasoning, and correctness verification. The iterative co-evolution process and explicit feedback loops are logically coherent and empirically well-supported.

**Weaknesses:**

### The Generalizability of the Paper is Limited

Despite claiming generalizability, the system is only evaluated using physics Olympiad problems derived from a single benchmark (HiPhO). No evidence is presented for transfer to related domains (such as mathematics Olympiads, chemistry Olympiads, or general science Q&A), nor are conceptual or empirical justifications provided for broader applicability beyond physics. I **would like to see evaluations targeting large-model agents on real scientific research problems** (even if sometimes limited to the physics domain, that would still be a significant breakthrough).

Moreover, the experiments are heavily designed around the structure of the HiPhO Olympiad (including structured JSON extraction for diagrams that conform to this benchmark’s conventions), creating a risk of benchmark-specific optimization. The paper could be improved by incorporating cross-benchmark generalization tests or, at the very least, providing deeper error analyses on out-of-distribution tasks/problems.

---

### Lack of Fairness in Experimental Comparison

In fairness comparisons, the paper uses a unified evaluation benchmark and scoring standard, ensuring result comparability. However, the experiments do not disclose hardware configurations for each method during runtime, total reasoning time, overall GPU usage, or token costs—only a single example’s relative consumption ratio is provided in the hyperparameter analysis. Given that the proposed method is a multi-stage, multi-round generation-and-verification system, **its inference overhead is clearly higher than that of single models and other baselines**. It is recommended that the authors supplement the final version with resource consumption comparisons for each method to allow readers to fully assess the real-world application cost. **If the strong results are not achieved under equal token or time consumption, they are meaningless and unacceptable.** If resources can be used without limits, then comparing with a single agent or single LLM is pointless—frequent agent calls will be hard to inspire either product development or academic research. (I acknowledge that many LLM-Agent papers have this issue, but it is still something that must be carefully considered.)

---

### Many Large-Model Prompts May Have Been Generated by LLMs

In the appendix, I noticed numerous prompts that, when tested on GPTZero, are highly likely to have been generated by an LLM. Extremely long prompts can cause the model’s attention to collapse, and I am not certain that Markdown is the best choice for prompt formatting. I cannot definitively confirm the prompts were LLM-generated, but the design philosophy and generation process still need to be described in detail in the paper. After all, if each task requires such an extensive prompt design process, usability will be very poor.

**Questions:**

1. It is necessary to clarify whether the potential research significance of your proposed method lies primarily in advancing the general development of large language models (LLMs), in contributing specifically to the "LLM for Science" direction, or in facilitating agent-based application development. At present, what I observe is essentially a finely crafted problem construction for a specific scenario. An adequate response should explain how your study can provide broader insights — for example, how it could inspire rapid design strategies when facing novel and challenging problems in future large models.

2. Please provide quantitative details regarding the computational overhead (in terms of both time and token usage) incurred by your proposed method. In typical use cases, how many times greater is the overhead compared to the Single Model approach illustrated in Figure 1? How does the time consumption compare under these conditions?

3. For *PhysicsMinions*, how is the decision made to completely reinitialize (i.e., discard and regenerate) a solution instead of continuing iterative refinement? Is this decision based on a fixed threshold, or is it determined adaptively?

The review scores may vary depending on the quality of your responses.

**Details Of Ethics Concerns:**

I observed extensive and direct use of the “Minions” character design and name, which are copyrighted and trademarked by Universal Pictures/Illumination Entertainment. I hope the author will carefully consider whether this may pose unnecessary copyright risks.

---

> ### Author Response · Authors · 2025-11-22
>
> Thank you for your thoughtful review. We sincerely appreciate your recognition of the strengths of our work, including its clear presentation, well-motivated design, and the historic milestone it achieves for the open-source benchmark. We will revise the paper accordingly.
>
> ---
>
> **Comment 1.** No evidence is presented for transfer to related domains (such as mathematics Olympiads, chemistry Olympiads, or general science Q&A), nor are conceptual or empirical justifications provided for broader applicability beyond physics. I would like to see evaluations targeting large-model agents on real scientific research problems.
>
> **Response 1.** Thank you for the valuable suggestion. Physics Olympiads offer an ideal testbed due to their demanding multimodal reasoning and strict physical consistency, which our PhysicsMinions system is specifically designed to address through components like a reflection-verification enhanced image-reading system (Visual Studio) and a general-specialized integrated dual-verifier system (Review Studio).
>
> As suggested, we further conduct additional experiments on broader domains, including **AIME 2025** (mathematics) and **SFE-Astronomy/Earth** (science QA) [1], as shown in Table R1, where our system achieves consistent performance improvements. Notably, on frontier physics research tasks (**CritPt**) [2], our system raised accuracy from 40% to 100% on Checkpoint 1 and enabled breakthrough performance on Checkpoint 2 with a partial answer, improving from 0% to 60% and highlighting its potential for scientific research applications. These results are now included in **Section 6** with detailed illustrations.
>
> **Table R1. Accuracy improvements across other disciplines and physics research tasks.**
> |Benchmark|AIME 2025|SFE-Astronomy|SFE-Earth|CritPt-Checkpoint 1|CritPt-Checkpoint 2 w/o Answer|CritPt-Checkpoint 2 w/ Answer|
> -|-|-|-|-|-|-
> |Gemini-2.5-Flash-Thinking|78.3%|24.3%|32.0%|40.0%|0.0%|0.0%
> |+ PhysicsMinions|93.3%|43.7%|46.9%|100.0%|20.0%|60.0%
> |*Improvement*|+15.0%|+19.4%|+14.9%|+60.0%|+20.0%|+60.0%
>
> [1] Scientists' First Exam: Probing Cognitive Abilities of MLLM via Perception, Understanding, and Reasoning. arXiv 2506.
>
> [2] Probing the Critical Point (CritPt) of AI Reasoning: a Frontier Physics Research Benchmark. arXiv 2509.
>
> ---
>
> **Comment 2.** The experiments do not disclose hardware configurations for each method during runtime, total reasoning time, overall GPU usage, or token costs. Given that the proposed method is a multi-stage, multi-round generation-and-verification system, its inference overhead is clearly higher than that of single models and other baselines. It is recommended that the authors supplement the final version with resource consumption comparisons for each method to allow readers to fully assess the real-world application cost.
>
> **Response 2.** Thank you for the constructive suggestion. We agree that evaluating resource usage is important for real-world applicability. While our coevolutionary multi-agent system introduces additional inference overhead compared to single-model baselines, it offers a unique advantage: it consistently **surpasses the performance ceiling of a standalone model** through introspection-verification cycles, and is fully compatible with any backbone model, including smaller models like Qwen2.5VL-32B-Instruct, which improves from bronze-level (9.9) to silver-level (12.4) on IPhO 2025 when enhanced by our system.
>
> As suggested, we conducted a comparison under **equal resource consumption**. On IPhO 2025, the average per-problem runtime for Intern-S1 is 180 seconds with 10,227 tokens, while for our system, it is 500 seconds with 89,482 tokens. This corresponds to a runtime ratio of approximately **1:3** and a token usage ratio of about **1:9**. All experiments were conducted by calling APIs on a Linux server, and thus do not involve direct GPU usage on our side. Despite this, our system achieves **avg@3 = 20.8**, whereas the best-of-N result reaches only **17.1** under equal runtime (N = 9) and equal token budget (N = 27), showing limited improvement. This demonstrates that increasing sampling alone cannot overcome the performance limit of single models, whereas our system consistently reaches ~20 points even with a single sample. **We have included these discussions in Section 5.5.**
>
> Moreover, our framework enables **weaker models to perform comparably to stronger ones**. For example, Intern-S1 improves from 2 gold medals to 6, matching Gemini-2.5-Flash-Thinking, offering a **more lightweight and deployable solution under limited computational resources and deployment budgets**. This makes our system especially valuable in scenarios where closed-source models are impractical, and its effectiveness has been validated across multiple open-source models.

---

> ### Author Response · Authors · 2025-11-22
>
> **Comment 3.** (1) In the appendix, I noticed numerous prompts that, when tested on GPTZero, are highly likely to have been generated by an LLM. (2) Extremely long prompts can cause the model’s attention to collapse, and I am not certain that Markdown is the best choice for prompt formatting. (3) I cannot definitively confirm the prompts were LLM-generated, but the design philosophy and generation process still need to be described in detail in the paper. After all, if each task requires such an extensive prompt design process, usability will be very poor.
>
> **Response 3.** **(1)** We would like to clarify that all prompts were **initially designed by humans** and lightly polished by GPT-5. To be more rigorous, we have added explicit statements in **Appendix A and C** to clarify it.
>
> **(2)** Regarding prompt length, our experiments show that **32B‑scale model can reliably handle the current prompt size**. The prompts used in our system fall within a range of **[80, 1902] tokens**, well below the **16K context window** supported by the 32B model, so context‑collapse is not a concern in practice.
>
> **(3)** Regarding the concern about prompt design, we emphasize the importance of using domain-specific prompts for physics Olympiads, where multimodal understanding and physics reasoning are essential. Our **Visual Studio prompts** are designed to extract key features from typical physics figures, such as axes, units, trends, and data points in line plots or variable-labeled diagrams. The **Physics-Verifier prompts** guide checks ranging from coarse-grained validation, such as unit consistency and physical constants, to fine-grained reasoning, including implicit assumptions and physical laws. **These prompt designs enable reliable error detection and have been added in detail in Appendix C**.
> Importantly, the prompts generalize well across physics problems and require only minimal adaptation for other domains, such as replacing "physics" with "math." As shown in Table R1 (Response 1), our system transfers effectively to mathematics, astronomy, earth science, and physics research tasks, demonstrating both the usability and adaptability of our prompt design.
>
> ---
>
> **Comment 4.** It is necessary to clarify whether the potential research significance of your proposed method lies primarily in advancing the general development of large language models (LLMs), in contributing specifically to the "LLM for Science" direction, or in facilitating agent-based application development. At present, what I observe is essentially a finely crafted problem construction for a specific scenario. An adequate response should explain how your study can provide broader insights — for example, how it could inspire rapid design strategies when facing novel and challenging problems in future large models.
>
> **Response 4.** Thank you for raising this important point. Our work represents one of the first coevolutionary multi-modal multi-agent systems specifically designed for physics Olympiads, a challenging setting that demands both multimodal understanding and physics-grounded reasoning. We believe this contributes meaningfully to the broader "LLM for Science" direction by addressing domain-specific challenges often overlooked in general LLM benchmarks.
>
> Beyond physics Olympiads, we demonstrate that our system, with minimal adjustments, generalizes to other disciplines, including mathematics, astronomy, earth science, and even frontier physics research tasks, as shown in Table R1 and **Section 6**. This highlights strong potential for broad scientific applications.
>
> Looking ahead, we plan to extend our framework with **adaptive optimization** (e.g., [3,4]), enabling more scalable and robust deployment in diverse and complex domains. These design insights could inform future strategies for rapidly adapting large models to novel, high-stakes scientific tasks.
>
> [3] Prorefine: Inference-time prompt refinement with textual feedback. arXiv 2506.
>
> [4] PRL: Prompts from Reinforcement Learning. arXiv 2505.

---

> ### Author Response · Authors · 2025-11-22
>
> **Comment 5.** For PhysicsMinions, how is the decision made to completely reinitialize (i.e., discard and regenerate) a solution instead of continuing iterative refinement? Is this decision based on a fixed threshold, or is it determined adaptively?
>
> **Response 5.** Thank you for the thoughtful question. To the best of our knowledge, PhysicsMinions is among the first coevolutionary multi-modal multi-agent systems specifically designed for physics Olympiads, and we believe the decision logic is an important aspect of its architecture. Currently, solution regeneration is based on a **fixed threshold**: if a solution fails two consecutive verification-introspection cycles, it is discarded and reinitialized (as described in Section 3.1). This threshold was validated through ablation studies, as shown in **Figure 5c**, which demonstrates its effectiveness. In future work, we plan to explore **adaptive strategies** that dynamically adjust the threshold based on model strength and task complexity.
>
> ---
>
> **Comment 6.** I observed extensive and direct use of the “Minions” character design and name, which are copyrighted and trademarked by Universal Pictures/Illumination Entertainment. I hope the author will carefully consider whether this may pose unnecessary copyright risks.
>
> **Response 6.** Thank you for the kind reminder. Our use of the term "Minions" and the associated character design is purely illustrative, intended as a light-hearted analogy to represent specialized agents within a coevolutionary system. We do not use any copyrighted content for commercial purposes. Our use of the "Minions" reference is similar in spirit to other research works that borrow from pop culture to make ideas more accessible. For example, prior publications have used public figures or cartoon characters in academic contexts:
>
> [1] *Mambaout: Do we really need mamba for vision?* (CVPR 2025) uses the likeness of Kobe Bryant.
>
> [2] *Manganese/iron organic frame material derivatives for rapid catalytic oxidation of metronidazole: performance and mechanism* (Journal of Water Process Engineering, 2024) features Tom and Jerry.
>
> [3] *Polydopamine mediated modification of manganese oxide on melamine sponge for photothermocatalysis of gaseous formaldehyde* (Journal of Hazardous Materials, 2021) references SpongeBob.
>
> That said, we fully respect intellectual property rights, and if any concerns arise, we are fully prepared to **replace both the name and the icons** (e.g., renaming to *PhysicsSolvers*) to avoid potential infringement.

---

> ### Author Response · Authors · 2025-11-27
> **Follow-up on Reviewer Concerns**
>
> Dear Reviewer,
>
> Thank you once again for your thoughtful feedback and for acknowledging the strengths of our work. As suggested, we have conducted additional experiments to demonstrate the broader generality of our system across mathematics, astronomy, earth science, and frontier physics research. We would like to confirm whether our rebuttal has satisfactorily addressed your concerns or if there are any remaining points that may still require clarification. We remain ready to provide further explanation if needed.
>
> If you find our response helpful, a higher score would be greatly appreciated.

---

### Official Review · Reviewer_Px8e · 2025-11-13

**Soundness:** 2
**Presentation:** 2
**Contribution:** 1
**Rating:** 2
**Confidence:** 4

**Summary:**

The paper proposes the so-called Physics Minions workflow, with three main components: the Visual Image studio for image description and measurements, the Logic studio for solutions, and the Review studio for verification. Experiments show that the proposed workflow improve performance of multiple LLMs on multiple physics olympaids, boosting open-source LLM performance to gold-medal-level. Authors also conducted further Pass@K and ablation experiments to validate design choices. Overall, authors claim the main improvement to be with respect to the image dealing part and the reviewing part.

**Strengths:**

The experiments are conducted on the IPhO bench, showing descent improvement brought about by the two major components (i.e. improvement from image measurement and reviewing component).

**Weaknesses:**

The most significant flaw of this paper is its lack in contribution: the two proposed components have been known facts. The experience and intuition that, accurate image measurement and self-refinement can improve physics reasoning capability of Large Language Models has been proposed and validated in previous work. For example, [1] shows methods like BoN and self-refinement works well on physics problems; [2] shows images are important for some physics olympiad problems. Even similar, [3] shows that provided with an image expert tool and a review expert tool, state-of-the-art LLMs can perform quite well on physics olympiad. Some figures in this submission also resemble those in [3]: this similarity **at least** shows that "accurate measurement and reviewing experts can boost performance in Physics Olympiad" has been discussed in previous work.

The authors have indeed conducted experiments to test many LLMs' performance on HiPhO bench, which is consisted of multiple physics olympiads. However, it is HiPhO bench and its evaluation methods that made this evaluation possible, not this submission.

Overall, this submission lacks novelty and contribution.


Referenced papers in this review:

[1] Improving Physics Reasoning in Large Language Models Using Mixture of
Refinement Agents, arXiv 2412.

[2] SeePhys: Does Seeing Help Thinking? -- Benchmarking Vision-Based Physics Reasoning, NeurIPS 2025.

[3] Physics Supernova: AI Agent Matches Elite Gold Medalists at IPhO 2025, arXiv 2509.

**Questions:**

As discussed in your paper, the VLM+agent architecture is better for image captioning and measurement, even better than some chart analysis tools tested, except the WebPlotDigitizer. The WebPlotDigitizer involves manual calibration of points measured. Could this manual calibration done by VLMs?

---

> ### Author Response · Authors · 2025-11-22
>
> **Comment 1.** The most significant flaw of this paper is its lack in contribution: the two proposed components have been known facts [1,2,3].
>
> **Response 1.** We respectfully **disagree** with the contribution concern. Our main contribution lies in developing a **coevolutionary multi-modal multi-agent system** tailored for physics Olympiads that **achieves the first open-source gold medal on IPhO 2025** and **generalizes across seven latest physics Olympiads** for the first time. We recognize that accurate image reading and self-refinement are crucial for the coevolutionary multi-modal multi-agent system in physics Olympiads. Therefore, we specifically design a reflection-verification enhanced image-reading system (Visual Studio) and a general-specialized integrated dual-verifier system (Review Studio). More specifically:
>
> **1. Physics Olympiad Requires More Than BoN or Self-Refinement:** As shown in Table 2, a single model yields only 2 golds; BoN or self-refine [1] improves that to 3 golds across 7 Olympiads. In contrast, our system achieves 6 golds, enabled by stable, cross-Olympiad gains. One key reason stems from our general-specialized integrated dual-verifier system (20.8 on IPhO 2025), which outperforms the single-reviewer setup used in self-refine (19.1–19.3), as shown in Figure 5b.
>
> **2. Our Reflection-Verification Enhanced Image-Reading System Enables Robust Image Understanding:** SeePhys [2] highlights the importance of diagrams but does *NOT* provide solutions to enhance visual reasoning. We address this via a reflection-verification Visual Studio. This consistently yields substantial gains on multimodal problems (Figure 3), with 10.0%-25.1% improvements as shown in Table R1, and also outperforms most chart analysis tools (Figure 7).
>
> **Table R1. Normalized score improvements across IPhO 2025 multimodal problems.**
> |Diagram Type|Illustration Figure|Variable Figure|Data Figure|
> |-|-|-|-|
> |Intern-S1|36.3%|66.6%|52.3%
> |+ PhysicsMinions|46.3%|84.4%|77.4%
> |*Improvement*|*+10.0%*|*+17.8%*|*+25.1%*
>
> **3. Our Work Is Contemporaneous, but Methodologically and Empirically Distinct from Physics-Supernova [3]:** While our work and [3] are contemporaneous (our ICLR abstract was submitted on **Sep. 3** and [3] was posted to arXiv on **Sep. 1**), our contributions are substantially different.
>
> - **Architecture:** [3] assumes the model itself possesses strong capabilities in image understanding and review, and their design for physics relies on external tools. In contrast, we recognize that most models, especially open-source ones, struggle with these capabilities. We therefore designed tailored modules:
>     - **(i) Visual Studio** incorporates reflection-verification to enhance multimodal perception across closed- and open-source models. [3], however, depends entirely on Gemini-2.5-Pro.
>     - **(ii) Review Studio** introduces a dual-verifier structure for coarse-to-fine error detection, which further supports iterative introspection.
>
>   Importantly, our system focuses on enhancing the model's internal capabilities through coevolution, rather than outsourcing reasoning to external tools.
>
> - **Generalization:** Our design enables consistent gains across **open- and closed-source models of various sizes**, validated on **seven diverse physics Olympiads**. In contrast, [3] focuses solely on IPhO 2025 and relies on the strong image and review capabilities of Gemini-2.5-Pro, limiting its generalizability to smaller or open-source models.
>
> - **Outcomes:** We demonstrate model evolution across weak-to-strong and strong-to-strong regimes, enabling Intern-S1 to achieve the **first open-source gold medal** on IPhO 2025 and elevating Gemini-2.5-Flash-Thinking to **all gold medals**. In contrast, [3] only explores the strong-to-strong setting and does not consider weak-to-strong transitions. Moreover, when testing Qwen2.5VL-32B-Instruct, we found that its one-shot pipeline frequently exceeds the 16K token limit imposed by the OpenRouter API, making it impractical for smaller models. By comparison, our system lifts Qwen2.5VL-32B-Instruct from 6 bronzes to **5 silvers and 2 bronzes**, highlighting broader applicability and scalability.
>
> **In sum, our work contributes a coevolutionary multimodal multi-agent system specifically designed for physics Olympiads, with generalization across models and breakthrough performance gains.** The direction of building such a coevolutionary agent system for physics competitions remains largely underexplored. Our reflection-verification enhanced image-reading system (Visual Studio) and a general-specialized integrated dual-verifier system (Review Studio) are novel and, to our knowledge, have not been previously validated; their innovation has been positively recognized by Reviewers YFh1 and KQyp. **We have added a detailed comparison with related work in the updated PDF (Section 5.4), and sincerely hope the reviewer can reassess the true value of our contributions.**

---

> ### Author Response · Authors · 2025-11-22
>
> **Comment 2.** The authors have indeed conducted experiments to test many LLMs' performance on HiPhO bench, which is consisted of multiple physics olympiads. However, it is HiPhO bench and its evaluation methods that made this evaluation possible, not this submission.
>
> **Response 2.** We respectfully disagree.
>
> - We do *NOT* claim HiPhO as a contribution; throughout the paper, we explicitly cite HiPhO as an existing benchmark (e.g., Lines 60, 85, 114, 259) and never present it as part of our novelty. Our contribution lies in the **system-level design: a coevolutionary multimodal multi-agent framework** featuring a reflection-verification enhanced image-reading system (Visual Studio) and a general-specialized integrated dual-verifier system (Review Studio). This system produces **consistent and substantial improvements across both open- and closed-source models and across seven distinct physics Olympiads**, and these performance improvements have been recognized by Reviewers YFh1, KQyp, and 9HJu.
>
> - Beyond HiPhO, **our system also generalizes to other disciplines and even physics research tasks**, including AIME 2025, SFE-Astronomy [4], SFE-Earth [4], and frontier physics research tasks (CritPt) [5], as shown in **Table R2** (with further details in **Section 6**). These results demonstrate the generality and broad applicability of our coevolutionary multimodal multi-agent design.
>
> **Table R2. Accuracy improvements across other disciplines and physics research tasks.**
> |Benchmark|AIME 2025|SFE-Astronomy|SFE-Earth|CritPt-Checkpoint 1|CritPt-Checkpoint 2 w/o Answer|CritPt-Checkpoint 2 w/ Answer|
> -|-|-|-|-|-|-
> |Gemini-2.5-Flash-Thinking|78.3%|24.3%|32.0%|40.0%|0.0%|0.0%
> |+ PhysicsMinions|93.3%|43.7%|46.9%|100.0%|20.0%|60.0%
> |*Improvement*|+15.0%|+19.4%|+14.9%|+60.0%|+20.0%|+60.0%
>
> [4] Scientists' First Exam: Probing Cognitive Abilities of MLLM via Perception, Understanding, and Reasoning. arXiv 2506.
>
> [5] Probing the Critical Point (CritPt) of AI Reasoning: a Frontier Physics Research Benchmark. arXiv 2509.
>
> ---
>
> **Comment 3.** As discussed in your paper, the VLM+agent architecture is better for image captioning and measurement, even better than some chart analysis tools tested, except the WebPlotDigitizer. The WebPlotDigitizer involves manual calibration of points measured. Could this manual calibration done by VLMs?
>
> **Response 3.** We appreciate the insightful question. While WebPlotDigitizer indeed achieves high-precision chart extraction, it relies on **manual calibration through human interaction**, such as marking axis origins, tick intervals, and value scales. In contrast, our focus is on **empowering the model itself** to perform accurate image measurement through **coevolving capabilities**, which enhances generality across models and domains, without depending on external tools. Using external tools (as done in [3]) or incorporating human-in-the-loop calibration are both promising directions for further improving performance. While the former has already shown benefits, the latter (using VLMs to automatically identify axis positions, tick spacing, and numerical ranges) remains largely unexplored. However, both directions are **beyond the scope of this work** and will be considered in future research.

---

> ### Author Response · Authors · 2025-11-27
> **Follow-up on Reviewer Concerns**
>
> Dear Reviewer,
>
> We have provided a comprehensive comparison with related work and revised the paper to clearly and explicitly articulate our contributions. We believe that our rebuttal has directly addressed the concerns raised. If there are still specific points that you consider unresolved, please let us know and we will clarify them timely.
>
> Thank you for your time and consideration.

---

### Meta-Review · Area_Chair_XCzP · 2025-12-29

**Summary:**

The paper presents a multimodal multi-agent system for solving physics-based reasoning problems, specifically those in the International Physics Olympiads, where the problems often have an image component (e.g., plots, diagrams, circuits) and a question text. The proposed modular multi-agent system comprises three modules: a visual studio for producing structured (json-based) text capturing details in the problem image, a logic studio that derives physics formulae for solving the problem, and a review studio that verifies the solution. All the three modules also contain agents that interprets the inputs and an introspector that verifies the outputs, including a logic-review loop where the logic and the review studio modules interact until they agree on the logic and solution. Experiments are provided on the latest physics Olympiads demonstrating gold medal level performance of the approach. Comparisons to alternative multi-agent models, including best-of-N, self-MoA, as well as various ablation studies are provided, exploring the benefits from each design choice.

**AC comments:**
The paper received mixed reviews, and the authors provided a strong rebuttal with additional experimental results that could potentially address some of the important concerns raised by the reviewers. However, considering the full set of concerns and authors responses, AC believes the paper represents an application of multi-agent solution/reflection ideas to the vision-and-language domain, where different multi-agent sub-modules operate on vision and language modalities separately. While the specific architecture demonstrates strong performance, its design motivation is unclear beyond existing actor–critic-style multi-agent models, which may explain the reviewers’ lack of enthusiasm regarding the technical contribution.

Additionally, the lack of the vision module’s participation in the full refinement process; limited details on performance under comparable computational budgets; insufficient discussion of image representation structure; lack of heterogeneity in the LLMs used; and absence of comparisons with feedback-based LLM approaches (e.g., Recursively Criticize and Improve, RECONCILE) render the contributions incremental. As such, AC believes the paper may fall below the novelty bar, despite demonstrating impressive performance on physics Olympiad problems.

**Reviewer Concerns:**

*Reviewer Px8e* acknowledges the improvements brought out by the image understanding and reviewing studios, however is concerned with the lack of major contributions the paper makes in the light of closely related prior works [1-3] that show similar contributions, including benefits from multi-agent aggregation and the use of expert tools for image understanding and review.

*Reviewer YFh1* is concerned with the lack of any conceptual contributions in the paper, that allows generalization of the approach beyond physics to other domains, including its design targeted to HiPhO Olympiad. A second concern is on the lack of transparency to the computational budget used in the varied experiments as well as in the design/generation of the prompts used.

*Reviewer KQyp* raised three concerns: i) lack of details on the computational budget, and results on similar budgets, ii) lack of heterogeneity in the deployed multi-LLMs, and iii) missing contrast to prior works that use multimodal LLMs for image understanding. The reviewer also raised an important point regarding detecting 'false consensus' that may be reached by the generation-verification loop.

*Reviewer 9HJu* raised two key concerns: i) the lack of any insights into how the proposed approach leads to better understanding of images/figures, and this extracted image information being fixed in the refinement process (reflection loop is only around the logic and review modules and not including the visual studio), and ii) lack of details on the computational budget/LLM calls between baselines and the proposed model.

**Reviewer Scores:**

**Reviewer Px8e:** Authors provide a strong rebuttal in response to the criticisms raised by this reviewer. There are two important take aways from the responses: i) that self-aggregation methods such as self-refinement and best-of-N that the reviewer points to have already been experimented with by the authors and demonstrate inferior results compared to Physics Minions, with their introspection loop leading to better image understanding (than say SeePhys) and ii) Physics-SuperNova [3] is a contemporaneous work that need not be compared to in this submission. The authors also present additional experiments demonstrating the generalizability of the approach to problems beyond physics, including AIME, astronomy, etc.

[*AC's thoughts on the response*] While the authors have technically addressed the questions that the reviewer is asking for, AC thinks the concerns are beyond the questions themselves, and is on the contribution that the presented approach brings. Prior works, including the ones that the reviewer is pointing out already shows homogeneous multi-agent feedback loops produce improved results, and such feedback based loops have been proposed earlier (e.g., methods such as Language Models can Solve Computer Tasks, Kim et al. 2023). Thus, while the reviewer may acknowledge the additional numerical results and would have raised the score, AC thinks it would still be below the bar.

**Reviewer YFh1** Authors provide a point-by-point response to the concerns. Specifically, i) additional experimental results are provided validating the efficacy of the model on non-physics problems demonstrating significant improvements, ii) an approximately balanced computational study against alternatives showing its not the computations, rather the model architecture that leads to the improved results, and iii) details regarding how the prompts were designed: initial human prompts refined by LLMs.

[*AC's thoughts on the response*] AC thinks the additional experiments on non-physics datasets could adequately address the reviewer's concerns regarding generalizability of the approach. AC also thinks that the experiments under similar computational budgets make sense and the results are reasonable and demonstrates the promise of the specific architecture. Thus, AC thinks the authors responses could have addressed the reviewer's concerns and the reviewer may have raised the score to 'marginally above acceptance'.

**Reviewer KQyp:** Responding to the question, authors claim that heterogeneous models lead to better computations and performance, while providing a limited study showing some promise, however these results need further analysis. As for the missing related works, authors include additional results, showing that their reflection-verification method outperforms multimodal LLMs in chart understanding. Authors also acknowledge that strong verification models are necessary for avoiding 'false consensus'.

[*AC's thoughts on the response*] Although the author's responses to the questions on heterogenous models and 'false consensus' could have been better, AC thinks the reviewer would have sustained their score at 'marginally above acceptance'.

**Reviewer 9HJu:** Authors emphasize that the visual studio module incorporates a reflection-verification loop that enhances the visual details extracted from the images, and this module when incorporated into the rest of the pipeline leads to state-of-the-art performance. On the second concern -- regarding the number of LLM calls -- authors remind that using more LLM calls do not lead to better results, instead their specific architecture is what matters.

[*AC's thoughts on the response*] The reviewer pointed out some important architectural issues, and the authors' response was not sufficient to provide insights into why it has to be so or if it could have been better. The issue regarding LLM calls appear to be reasonably addressed. AC thinks the reviewer may have sustained the score at 'below acceptance'.

---

### Decision · Program_Chairs · 2026-01-26

Reject